# Fingerprinting Deep Neural Networks for Ownership Protection: An Analytical Approach

**Guang Yang**[1]**, Ziye Geng**[2]**, Yihang Chen**[2]**, and Changqing Luo**[2]
[1]Virginia Commonwealth University, [2]University of Houston
`yangg2@vcu.edu`, `{zgeng2, ychen165, cluo3}@uh.edu`

## Abstract

Adversarial-example-based fingerprinting approaches, which leverage the decision boundary characteristics of deep neural networks (DNNs) to craft fingerprints, has proven effective for protecting model ownership. However, a fundamental challenge remains unresolved: how far a fingerprint should be placed from the decision boundary to simultaneously satisfy two essential properties—robustness and uniqueness—required for effective and reliable ownership protection. Despite the importance of the fingerprint-to-boundary distance, existing works offer no theoretical solution and instead rely on empirical heuristics to determine it, which may lead to violations of either robustness or uniqueness properties.

We propose AnaFP, an analytical fingerprinting scheme that constructs fingerprints under theoretical guidance. Specifically, we formulate the fingerprint generation task as the problem of controlling the fingerprint-to-boundary distance through a tunable stretch factor. To ensure both robustness and uniqueness, we mathematically formalize these properties that determine the lower and upper bounds of the stretch factor. These bounds jointly define an admissible interval within which the stretch factor must lie, thereby establishing a theoretical connection between the two constraints and the fingerprint-to-boundary distance. To enable practical fingerprint generation, we approximate the original (infinite) sets of pirated and independently trained models using two finite surrogate model pools and employ a quantile-based relaxation strategy to relax the derived bounds. Particularly, due to the circular dependency between the lower bound and the stretch factor, we apply a grid search strategy over the admissible interval to determine the most feasible stretch factor. Extensive experimental results demonstrate that AnaFP consistently outperforms prior methods, achieving effective and reliable ownership verification across diverse model architectures and model modification attacks.

## 1 Introduction

Deep neural networks (DNNs) are increasingly vulnerable to model piracy in real-world deployments (Ren et al., 2023; Choi et al., 2025; Yao et al., 2025b). Adversaries may steal a model, apply performance-preserving model modification attacks to evade detection, and distribute these pirated ones as black-box services, thereby profiting from public usage while concealing the models' internal details. This emerging threat raises serious concerns about the intellectual property (IP) protection of DNN models. So far, extensive research has explored ownership protection techniques, such as watermarking (Fan et al., 2019; Shafieinejad et al., 2021; Li et al., 2024; Choi et al., 2025; Yao et al., 2025a) and fingerprinting (Chen et al., 2022; Pan et al., 2022; Liu & Zhong, 2024; Godinot et al., 2025). While watermarking involves embedding identifiable patterns into DNN models—potentially introducing new security vulnerabilities (Wang et al., 2021a; Xu et al., 2024), fingerprinting offers a non-intrusive alternative by leveraging the model's intrinsic characteristics to generate unique, verifiable fingerprints. This non-intrusive nature has made fingerprinting an increasingly appealing ownership protection approach.

To date, a broad spectrum of model fingerprinting schemes has been proposed (Cao et al., 2021; Guan et al., 2022; Wang et al., 2021a; Xu et al., 2024; Zhao et al., 2024a; You et al., 2024; Ren et al., 2023; Yin et al., 2022; Yang & Lai, 2023; Peng et al., 2022; Liu & Zhong, 2024; Godinot et al., 2025).

These schemes involve two phases: fingerprint generation and ownership verification (Lukas et al., 2021; Yang et al., 2022). During the fingerprint generation phase, the inherent characteristics of a protected DNN model are leveraged to carefully craft fingerprints that elicit model-specific responses. In the subsequent ownership verification phase, these fingerprints are used to assess whether a suspect model is a pirated version of the protected model. The goal is to reliably identify pirated models derived from the protected one, while avoiding false attribution of independently trained models. To ensure this, effective fingerprints must exhibit two essential properties: 1) robustness—the ability to withstand performance-preserving model modifications, and 2) uniqueness—the capability to distinguish the protected model from other independently trained models (Pan et al., 2022).

A prominent line of work among existing model fingerprinting schemes builds on the key observation that decision boundaries are highly model-specific, even across models trained on the same dataset (Somepalli et al., 2022). This motivates adversarial-example-based fingerprinting approaches that leverage the unique decision boundary characteristics of a model to craft fingerprints (Cao et al., 2021; Wang et al., 2021a; Lukas et al., 2021; Yang & Lai, 2023; Peng et al., 2022; Zhao et al., 2020). More specifically, fingerprints are crafted by applying minimal perturbations to input samples, pushing them across the decision boundary of the protected model, akin to adversarial example generation. Due to the inherent differences in decision boundaries, such perturbations typically induce prediction changes in the protected model but leave independently trained models largely unaffected (Szegedy et al., 2014; Zhao et al., 2020). This resulting prediction asymmetry enables the construction of distinctive and reliable fingerprints for ownership verification (Zhao et al., 2020).

However, a fundamental challenge in adversarial-example-based fingerprinting is: how far should a fingerprint be placed from the decision boundary to simultaneously satisfy the requirements of robustness and uniqueness? Ensuring uniqueness requires placing fingerprints close to decision boundaries, while enhancing robustness requires positioning fingerprints far from the boundaries. Recent works (Cao et al., 2021; Liu & Zhong, 2024) have employed heuristic strategies to empirically select the fingerprint-to-boundary distance. However, such approaches lack theoretical guidance, potentially resulting in fingerprints that violate the two desired properties. Consequently, how to theoretically determine the fingerprint-to-boundary distance that simultaneously satisfies both uniqueness and robustness properties remains an open issue.

In this paper, we propose **AnaFP**, an **Ana**lytical adversarial-example-based **F**inger**P**rinting scheme that crafts fingerprints under theoretical guidance. Specifically, we first identify high-confidence samples from the dataset used to train the protected model as anchors for fingerprint generation. For each anchor, we compute a minimal perturbation that induces a prediction change in the protected model, thereby ensuring strong uniqueness. To further improve robustness, we introduce a stretch factor that scales the perturbation, controlling the fingerprint's distance from the decision boundary. To ensure both robustness and uniqueness, we mathematically formalize the robustness and uniqueness constraints and theoretically derive the lower and upper bounds of the stretch factor accordingly. These bounds define an admissible interval within which the stretch factor must lie, thus establishing a theoretical relationship between the two constraints and the fingerprint-to-boundary distance.

Although this theoretical relationship offers strong theoretical guarantees, its practical implementation poses several challenges. First, computing the bounds requires worst-case parameter estimation over all possible pirated and independently trained models—two theoretically infinite sets. Thus, we approximate the two sets using two surrogate model pools. Second, using the worst-case estimates often leads to overly conservative bounds, reducing the feasibility of fingerprint construction. To mitigate this, we employ a quantile-based relaxation mechanism to relax the two bounds through quantile-based parameter estimation. Third, there exists a circular dependency between the lower bound and the stretch factor. We address this by applying a grid search strategy over the admissible interval to determine the most feasible value for the stretch factor. We conduct extensive experiments across diverse model architectures and datasets, and experimental results demonstrate that AnaFP consistently achieves superior ownership verification performance, even under performance-preserving model modification attacks, outperforming existing adversarial-example-based fingerprinting approaches.

## 2 BACKGROUND

### 2.1 ADVERSARIAL EXAMPLES

Adversarial examples are carefully crafted from samples in a clean dataset to induce a target model to make incorrect predictions (Szegedy et al., 2014; Goodfellow et al., 2015). Given a model $f$ and an input-label pair $(x, y)$ in a clean dataset $D$, an adversarial example $\hat{x} = x + \delta$ can be obtained by solving:

$$\min \|x - \hat{x}\|, \quad \text{s.t.} \quad f(\hat{x}) \neq y, \tag{1}$$

where $\|\cdot\|$ denotes a distance metric (e.g., $\ell_2$ or $\ell_\infty$ norm), and the perturbation $\delta$ is constrained to ensure imperceptibility. By inducing misclassifications with minimal perturbations, adversarial examples reveal the unique characteristics of a model's decision boundary. This property makes them particularly effective for constructing fingerprints that capture model-specific behaviors used for ownership verification.

### 2.2 MODEL MODIFICATION ATTACKS

To circumvent ownership verification, adversaries usually launch performance-preserving model modification attacks that alter a pirated model's decision boundary while maintaining predictive accuracy. Common techniques include fine-tuning (Zhuang et al., 2020), pruning (Li et al., 2017), knowledge distillation (KD) (Hinton et al., 2015), and adversarial training (Zhao et al., 2024b). Specifically, fine-tuning adjusts the weights of a pirated model through additional model training; pruning eliminates less significant weights or neurons of a pirated model to reshape its structure; KD trains a new model with a different structure and parameters to replicate the behavior of a pirated model; and adversarial training updates model weights using a mixture of clean data and adversarial examples, thereby posing a unique threat to adversarial-example-based fingerprinting methods. Even worse, adversaries may combine multiple techniques, such as pruning followed by fine-tuning, to induce substantial shifts in the decision boundary, thereby increasing the likelihood of invalidating fingerprints.

## 3 PROBLEM FORMULATION

We consider a typical simplified model finger-printing scenario with two parties: a model owner and an attacker, as shown in Figure 1. Specifically, the model owner trains a DNN model $P$ and deploys it as a service. The attacker acquires an unauthorized copy of $P$, applies performance-preserving model modifications to it, and subsequently redistributes the pirated model in a black-box manner (e.g., via API access). Let $\mathcal{V}_P$ denote the set of pirated models derived from $P$. To safeguard the intellectual property of $P$, the model

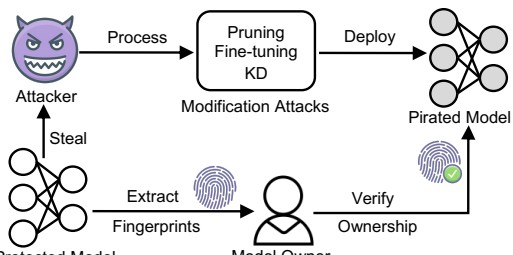

Figure 1: A typical model fingerprinting scenario.

owner leverages the adversarial example technique to produce a set of fingerprints $\mathcal{F}$ for $P$. Once identifying a suspect model $S$, the owner initiates ownership verification by querying $S$ using fingerprints in $\mathcal{F}$. The goal is to determine whether $S$ belongs to the pirated model set $\mathcal{V}_P$ or the independent model set $\mathcal{I}_P$ that includes models independently trained from scratch without any access to $P$.

To enable reliable verification, the *fingerprint set* $\mathcal{F} = \{(x_i^\star, y_i^\star) | 1 \leq i \leq N_f\}$, where $N_f$ is the number of fingerprints, $x_i^\star \in \mathcal{X} \in \mathbb{R}^{d_1 \times \ldots \times d_n}$ is the input of fingerprint $i$, and $y_i^\star \in \mathcal{Y} = \{1, \ldots, K\}$ is the corresponding target label, is constructed with the goal of satisfying the following properties:

- **Robustness:** Any fingerprint $(x_i^\star, y_i^\star) \in \mathcal{F}$ remains effective against performance-preserving modification attacks, i.e., $P'(x_i^\star) = y_i^\star$, for $\forall P' \in \mathcal{V}_P$.

- **Uniqueness:** Any fingerprint $(x_i^\star, y_i^\star) \in \mathcal{F}$ is unlikely to be correctly predicted by any independently trained models, i.e., $I(x_i^\star) \neq y_i^\star$, for $\forall I \in \mathcal{I}_P$.

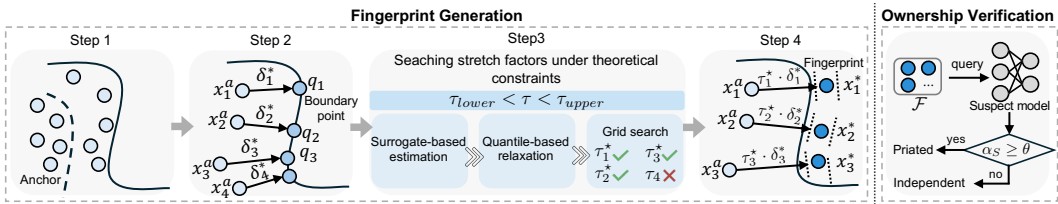

Figure 2: The pipeline of fingerprint generation and ownership verification.

# 4 THE DESIGN OF ANAFP

We present AnaFP, an analytical fingerprinting scheme that mathematically formalizes the robustness and uniqueness properties, constructs practical fingerprints satisfying relaxed versions of these constraints, and enables ownership verification using these fingerprints. As illustrated in Figure 2, AnaFP comprises two phases: fingerprint generation and ownership verification. Particularly, the fingerprint generation phase is composed of four main steps. The first step selects high-confidence samples from a clean dataset as anchors, ensuring that independently trained models predict their original labels with high probability. The second step computes a minimal perturbation for each anchor such that the protected model, when evaluated on the perturbed anchor, outputs a different label from that of the original anchor. The third step calculates a stretch factor for each anchor. To this end, an admissible interval for a stretch factor is derived by first mathematically formulating robustness and uniqueness constraints and then relaxing the formulated ones based on surrogate-based parameter estimation and quantile-based relaxation. With this interval, a feasible stretch factor is determined by exploiting a grid search strategy. The fourth step generates fingerprints by first scaling the perturbation with the obtained stretch factor and then applying it to its corresponding anchor. All resulting fingerprints form the final fingerprint set that will be used for ownership verification. In the following, we detail the four steps of fingerprint generation and the ownership verification protocol.

## 4.1 STEP 1: SELECTING HIGH-CONFIDENCE ANCHORS

In the first step, we identify high-confidence samples from the training dataset used to train the protected model. These input-label pairs, referred to as anchors, are selected based on the model's confidence in correctly predicting their labels. To quantify the confidence, we define the logit margin of a sample $(x, y) \in D$ with respect to the protected model $P$ as $g_P(x) = s_{P,y}(x) - \max_{k \neq y} s_{P,k}(x)$, where $s_{P,k}(x)$ is the logit (i.e., the pre-softmax outputs) of class $k$ outputted by $P$ for input $x$.

We construct the anchor set $\mathcal{A}$ by selecting samples whose logit margins exceed a predefined threshold $m_{\text{anchor}}$, i.e., $\mathcal{A} = \{(x^a, y) \in D \mid g_P(x^a) \geq m_{\text{anchor}}\}$. Intuitively, high-margin samples exhibit class-distinctive features that are reliably captured across independently trained models. As such, using these samples as anchors helps enhance the uniqueness of the resulting fingerprints.

## 4.2 STEP 2: COMPUTING MINIMAL DECISION-ALTERING PERTURBATIONS

In the second step, we compute a minimal perturbation for each anchor and apply it to produce a perturbed anchor that causes the protected model to change its prediction. Given an anchor $(x^a, y)$, we formulate a decision-altering perturbation minimization problem as

$$\delta^* = \arg \min_{\delta} \|\delta\|_2, \quad \text{s.t.} \quad P(x^a + \delta) \neq y, \tag{2}$$

where $\delta^*$ is the minimal perturbation for an anchor $(x^a, y) \in \mathcal{A}$, and $\|\cdot\|_2$ is the $\ell_2$ norm.

To solve the formulated problem, we employ the Carlini & Wagner $\ell_2$ attack (C&W-$\ell_2$) (Carlini & Wagner, 2017) to efficiently compute $\delta^*$. The resulting perturbed input is $q = x^a + \delta^*$, which we refer to as the *boundary point*, as it resides on a decision boundary of the protected model. This is the closest point to the anchor $(x^a, y)$ at which the protected model $P$ alters its prediction to a different label.

### 4.3 STEP 3: SEARCHING STRETCH FACTORS UNDER THEORETICAL CONSTRAINTS

Since boundary points are located directly on the decision boundary, they are highly vulnerable to boundary shifts caused by model modifications, often resulting in changes in model predictions. To improve robustness, we stretch the perturbation $\delta^*$ along its direction and apply it to the anchor to generate a fingerprint positioned farther from the boundary. Specifically, given an anchor $(x^a, y)$, the corresponding fingerprint is defined as $(x^\star = x^a + \tau\delta^*, y^\star = P(x^\star))$, where $\tau > 1$ is a stretch factor that controls the distance of the fingerprint from the decision boundary.

A central challenge lies in selecting an appropriate stretch factor $\tau$, as the resulting fingerprint needs to satisfy both robustness and uniqueness properties. To mathematically formalize these requirements, we first define a lower bound $\tau_{\text{lower}}$ and an upper bound $\tau_{\text{upper}}$ with respect to $\tau$, corresponding to the robustness and uniqueness constraints, respectively: the robustness constraint requires $\tau > \tau_{\text{lower}}$, while the uniqueness constraint requires $\tau < \tau_{\text{upper}}$. Then, we derive a theoretical constraint that defines the admissible interval of the stretch factor $\tau$:

$$1 + \frac{2\,\epsilon_{\text{logit}}}{c_g\,\|\delta^*\|} = \tau_{\text{lower}} < \tau < \tau_{\text{upper}} = \frac{m_{\min}}{2L_{\text{uniq}}\,\|\delta^*\|}, \tag{3}$$

where $c_g = \|\nabla g_P(q)\|_2$ is the norm of the gradient of the logit margin function $g_P$ at the boundary point $q = x^a + \delta^*$, as well as $m_{\min}$, $L_{\text{uniq}}$, and $\epsilon_{\text{logit}}$ are defined as follows:

- $m_{\min}$: a lower bound on the logit margin of independently trained models at the anchor $(x^a, y)$: $s_{I,y}(x^a) - \max_{k \neq y} s_{I,k}(x^a) \geq m_{\min}$, for $\forall k \in \mathcal{Y}, \forall I \in \mathcal{I}_P$.

- $L_{\text{uniq}}$: an upper bound on the local Lipschitz constant of independently trained models in the region between $x^a$ and $x^\star$, such that $\frac{\|s_I(x^\star) - s_I(x^a)\|_2}{\|x^\star - x^a\|_2} \leq L_{\text{uniq}}$, for $\forall I \in \mathcal{I}_P$, and $s_I(\cdot)$ is the logit vector of an independently trained model $I$.

- $\epsilon_{\text{logit}}$: an upper bound on the logit shift at $x^\star$ due to the performance-preserving model modification attack on a protected model $P$: $|s_{P',k}(x^\star) - s_{P,k}(x^\star)| \leq \epsilon_{\text{logit}}, \forall k \in \mathcal{Y}, \forall P' \in \mathcal{V}_P$.

The detailed derivation of Equation (3) is provided in Appendix A.

#### 4.3.1 PARAMETER ESTIMATION AND RELAXATION

**Surrogate-based estimation.** $m_{\min}$, $L_{\text{uniq}}$, and $\epsilon_{\text{logit}}$ are defined over $\mathcal{V}_P$ and $\mathcal{I}_P$, which, however, contain an infinite number of models respectively. Consequently, it is impossible to compute them directly. To address this issue, a common approach is to introduce two finite surrogate model pools to approximate the original sets (Li et al., 2021; Pan et al., 2022): a surrogate pirated pool $\mathcal{V}_P^s$ and a surrogate independent pool $\mathcal{I}_P^s$. $\mathcal{V}_P^s$ contains the protected model $P$ and its pirated variants, while $\mathcal{I}_P^s$ is composed of independently trained models.

Then, each parameter is estimated over the corresponding surrogate pool: $m_{\min}$ is estimated as the minimum margin over $\mathcal{I}_P^s$, $L_{\text{uniq}}$ is estimated as the maximum local Lipschitz constant over $\mathcal{I}_P^s$, and $\epsilon_{\text{logit}}$ is estimated as the maximum logit shift over $\mathcal{V}_P^s$. While these finite surrogate pools serve as practical approximations of the infinite sets $\mathcal{I}_P$ and $\mathcal{V}_P$, this substitution inevitably relaxes the original theoretical constraints and introduces approximation error. Although such error is theoretically intractable and dependent on the choice of surrogate models, experimental results in Section 5.2.1 demonstrate that AnaFP remains robust to variations in pool size and diversity, consistently yielding stable verification performance.

**Quantile-based relaxation.** Directly adopting the most conservative estimates, i.e., the minimum value of $m_{\min}$ and the maximum values of $L_{\text{uniq}}$ and $\epsilon_{\text{logit}}$, may lead to overly strict lower and upper bounds on $\tau$, potentially resulting in a situation where no fingerprints simultaneously satisfy the constraints for robustness and uniqueness. To address this issue, we employ a quantile-based relaxation strategy. Instead of using extreme values, we estimate each parameter based on its empirical quantile distribution over the surrogate pools: $m_{\min}$ is set to the $q_{\text{margin}}$-quantile of the logit margins computed over $\mathcal{I}_P^s$, $L_{\text{uniq}}$ is set to the $q_{\text{lip}}$-quantile of local Lipschitz constants over $\mathcal{I}_P^s$, and $\epsilon_{\text{logit}}$ is set to the $q_{\text{eps}}$-quantile of logit shifts measured over $\mathcal{V}_P^s$. In this way, we relax the lower and upper bounds on $\tau$, balancing theoretic rigor with practical feasibility.

### 4.3.2 THE GRID SEARCH STRATEGY FOR FINDING A FEASIBLE $\tau$

Based on Equation (3), the stretch factor $\tau$ is constrained by a lower bound $\tau_{\text{lower}}$ and an upper bound $\tau_{\text{upper}}$. While $\tau_{\text{upper}}$ can be explicitly computed using the estimated parameters, $\tau_{\text{lower}}$ is defined implicitly as a function of $\tau$—since it depends on the logit shift at the point $x^\star = x_a + \tau\delta^*$. This circular dependency introduces a non-trivial challenge in directly solving for $\tau_{\text{lower}}$. As a result, it becomes necessary to search for a feasible value of $\tau$ that satisfies the constraint $\tau_{\text{lower}}(\tau) \leq \tau \leq \tau_{\text{upper}}$ and simultaneously achieves a balance between uniqueness and robustness in the resulting fingerprint.

To address this, we exploit a grid search strategy to determine the feasible $\tau$. Specifically, given that $\tau_{\text{lower}} = 1 + \frac{2\,\epsilon_{\text{logit}}}{c_g\,\|\delta^*\|} \geq 1$, we define the search interval as $(1, \tau_{\text{upper}}]$. We first construct a candidate set $\mathcal{T}$ by uniformly sampling stretch factors within this interval. For each $\tau \in \mathcal{T}$, we compute $\tau_{\text{lower}}(\tau)$ and retain those satisfying $\tau \geq \tau_{\text{lower}}(\tau)$ to form the feasible set $\mathcal{T}_{\text{feas}}$. If no valid $\tau$ exists within the search interval (i.e., $\mathcal{T}_{\text{feas}} = \emptyset$), the corresponding anchor is discarded, as it cannot yield a valid fingerprint under the relaxed constraints. This infeasibility may arise in two cases: (i) $\tau_{\text{upper}} < 1$, rendering the interval void, or (ii) $\tau_{\text{upper}} \geq 1$, but no candidate $\tau$ satisfies $\tau > \tau_{\text{lower}}(\tau)$. Finally, from the feasible set $\mathcal{T}_{\text{feas}}$, we select the most feasible stretch factor $\tau^\star$ by $\tau^\star = \arg\max_{\tau \in \mathcal{T}_{\text{feas}}} \min\left\{ \tau - \tau_{\text{lower}}(\tau),\ \tau_{\text{upper}} - \tau \right\}$. This selection maximizes the minimum slack to both bounds, offering the greatest possible tolerance to parameter-estimation errors.

## 4.4 STEP 4: CONSTRUCTING THE FINGERPRINT SET

After determining the most feasible stretch factor $\tau_i^\star$ for each retained anchor $x_i^a$, we proceed to generate the corresponding fingerprints. Specifically, we scale the minimal decision-altering perturbation $\delta_i^\star$ by $\tau_i^\star$, applying it to the anchor $x_i^a$, and then record the resulting label assigned by the protected model $P$. The final fingerprint set $\mathcal{F}$ is constructed as: $\mathcal{F} = \big\{ (x_i^\star, y_i^\star) \mid (x_i^\star, y_i^\star) = \big(x_i^a + \tau_i^\star\,\delta_i^*,\ P(x_i^a + \tau_i^\star\,\delta_i^*)\big)\big\}_{i=1}^{N_f}$. By utilizing multiple fingerprints, our proposed scheme mitigates the potential impact of approximation errors—those introduced during the parameter estimation and relaxation in Step 3—that may affect the verification performance of individual fingerprint, thereby enhancing the overall reliability of the ownership verification.

## 4.5 VERIFICATION PROTOCOL

Given a suspect model $S$, the model owner verifies ownership by querying it with fingerprints in $\mathcal{F} = \{(x_i^\star, y_i^\star)\}_{i=1}^{N_f}$. Specifically, for a fingerprint $(x_i^\star, y_i^\star) \in \mathcal{F}$, the model owner queries $S$ using $x_i^\star$ and compares the returned label $S(x_i^\star)$ with $y_i^\star$. If $S(x_i^\star) = y_i^\star$, the model owner counts it as a match. After querying all fingerprints, the model owner computes the matching rate $\alpha_S = \frac{1}{N_f}\sum_{i=1}^{N_f} \mathbf{1}\big[S(x_i^\star) = y_i^\star\big]$, where $\mathbf{1}[\cdot]$ is the indicator function. If $\alpha_S \geq \theta$, where $\theta \in [0, 1]$ is a decision threshold, the suspect model $S$ is classified as pirated; otherwise, it is deemed independent.

## 5 EXPERIMENTS

We conduct extensive experiments to evaluate the effectiveness of AnaFP across three representative types of deep neural networks: convolutional neural network (CNN) models trained on the CIFAR-10 dataset (Krizhevsky, 2009) and the CIFAR-100 dataset (Krizhevsky, 2009), multilayer perceptron (MLP) models trained on the MNIST dataset (LeCun et al., 2010), and graph neural network (GNN) models trained on the PROTEINS dataset (Borgwardt et al., 2005).

**Model construction.** For each task, we designate a protected model and construct two surrogate model sets as well as two testing model sets:

- **Protected model**: The protected model architectures for CNN, MLP, and GNN are ResNet-18, ResMLP, and GAT.

- **Surrogate model sets**: For each task, we construct a pirated model pool consisting of six models obtained via two simulated performance-preserving modifications of the protected model (fine-tuning and knowledge distillation), and an independent model pool including six models independently trained from scratch with two different architectures and random seeds.

- **Testing model sets**: For each task, we construct a pirated model set via modification attacks, including pruning, fine-tuning, knowledge distillation (KD), adversarial training (AT), N-finetune (injecting noise into weights followed by fine-tuning), and P-finetune (pruning followed by fine-tuning). Each type of model modification produces 20 variants using different random seeds, resulting in 120 pirated models in this set. Besides, we construct an independent model set, and the independently trained models are trained from scratch with varying architectures and seeds, without any access to the protected model and its variants. This set includes 120 independently trained models.

The two testing sets constitute the evaluation benchmark used to test whether AnaFP can effectively discriminate pirated models from independently trained ones. The models in the testing sets used for performance testing are all unseen/unknown during the fingerprint generation process and have no overlap with the models in the surrogate pools.

**Baselines.** We compare AnaFP with six representative model fingerprinting approaches: **UAP** (Peng et al., 2022), which generates universal adversarial perturbations as fingerprints; **IPGuard** (Cao et al., 2021), which probes near-boundary samples to capture model-specific behavior; **MarginFinger** (Liu & Zhong, 2024), which controls the distance between a fingerprint and the decision boundary to gain robustness; **AKH** (Godinot et al., 2025), a recently proposed fingerprinting method that uses the protected model's misclassified samples as fingerprints; **GMFIP** (Yan et al., 2025), a recently-proposed non-adversarial-example-based fingerprinting method that trains a generator to synthesize fingerprint samples; and **ADV-TRA** (Xu et al., 2024), a fingerprinting method that constructs adversarial trajectories traversing across multiple boundaries.

Table 1: AUCs achieved by different approaches across DNN models and datasets.

| Method | CNN (CIFAR-10) | CNN (CIFAR-100) | MLP (MNIST) | GNN (PROTEINS) |
|---|---|---|---|---|
| AnaFP (ours) | **0.957 ± 0.002** | **0.893 ± 0.005** | **0.963 ± 0.002** | **0.926 ± 0.005** |
| UAP | 0.850 ± 0.010 | 0.806 ± 0.021 | 0.906 ± 0.004 | – |
| IPGuard | 0.715 ± 0.075 | 0.725 ± 0.090 | 0.873 ± 0.018 | 0.636 ± 0.067 |
| MarginFinger | 0.671 ± 0.064 | 0.630 ± 0.072 | 0.653 ± 0.051 | – |
| AKH | 0.723 ± 0.016 | 0.802 ± 0.019 | 0.851 ± 0.013 | 0.854 ± 0.021 |
| ADV-TRA | 0.878 ± 0.012 | 0.850 ± 0.024 | 0.887 ± 0.005 | – |
| GMFIP | 0.814 ± 0.047 | 0.781 ± 0.075 | 0.892 ± 0.023 | – |

**Evaluation metric.** We adopt the Area Under the Receiver Operating Characteristic (ROC) Curve (AUC) as the primary metric. AUC measures the probability that a randomly selected pirated model exhibits a higher fingerprint matching rate than an independently trained model that is randomly selected, thereby quantifying the discriminative capability of the fingerprints across all possible decision thresholds. A higher AUC value indicates stronger discriminative capability, with an AUC of 1.0 representing perfect discrimination between pirated and independently trained models.

## 5.1 EFFECTIVENESS OF OUR DESIGN

**The discriminative capability of AnaFP.** To evaluate the discriminative capability of AnaFP, we conduct experiments across multiple types of DNN models (CNN, MLP, and GNN) trained on different datasets (CIFAR-10, CIFAR-100, MNIST, and PROTEINS). Note that UAP and MarginFinger are not evaluated on GNNs as they are designed for data with Euclidean structures, such as images or vectors, and cannot generalize to graph-structured (i.e., non-Euclidean) data.

All experiments were independently run five times to find the mean and standard deviation across the runs.

The experimental results are summarized in Table 1. Specifically, we observe that AnaFP consistently achieves the highest

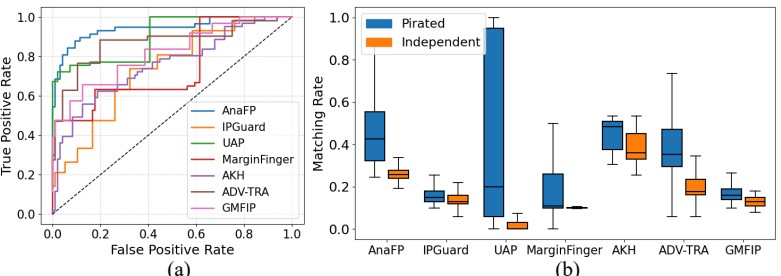

Figure 3: (a) The ROC curve and (b) the matching rate distribution.

Table 2: AUCs under various performance-preserving model modifications.

| Method | Pruning | Fine-tuning | KD | AT | N-finetune | P-finetune | Prune-KD |
|---|---|---|---|---|---|---|---|
| AnaFP (ours) | **1.000 ± 0.000** | **0.979 ± 0.008** | **0.756 ± 0.023** | **0.983 ± 0.015** | **0.978 ± 0.010** | **0.989 ± 0.007** | **0.689 ± 0.031** |
| UAP | **1.000 ± 0.000** | 0.868 ± 0.023 | 0.679 ± 0.013 | 0.783 ± 0.041 | 0.870 ± 0.021 | 0.876 ± 0.022 | 0.625 ± 0.026 |
| IPGuard | 0.999 ± 0.002 | 0.741 ± 0.113 | 0.559 ± 0.105 | 0.616 ± 0.026 | 0.679 ± 0.104 | 0.671 ± 0.106 | 0.596 ± 0.079 |
| MarginFinger | **1.000 ± 0.000** | 0.658 ± 0.119 | 0.554 ± 0.083 | 0.672 ± 0.087 | 0.586 ± 0.064 | 0.638 ± 0.115 | 0.543 ± 0.088 |
| AKH | 0.999 ± 0.001 | 0.848 ± 0.016 | 0.616 ± 0.054 | 0.627 ± 0.122 | 0.716 ± 0.039 | 0.701 ± 0.052 | 0.636 ± 0.047 |
| ADV-TRA | **1.000 ± 0.000** | 0.923 ± 0.010 | 0.660 ± 0.025 | 0.742 ± 0.036 | 0.895 ± 0.032 | 0.857 ± 0.013 | 0.633 ± 0.035 |
| GMFIP | 0.999 ± 0.001 | 0.779 ± 0.054 | 0.591 ± 0.035 | 0.711 ± 0.084 | 0.858 ± 0.031 | 0.819 ± 0.061 | 0.601 ± 0.068 |

AUCs across all evaluation settings, whereas the baselines exhibit substantial variability across different models and datasets. This observation underscores AnaFP's superiority in distinguishing pirated models from independently trained ones across diverse DNN model types and data modalities, demonstrating its effectiveness for ownership verification.

To further illustrate AnaFP's discriminative capability, we present both ROC curves and fingerprint matching rate distributions on a representative case with CNN models trained on CIFAR-10. The ROC curve plots the true positive rate (TPR) against the false positive rate (FPR) across varying verification thresholds, providing a comprehensive view of the fingerprint's discriminative ability. A curve closer to the top-left corner indicates stronger separability between pirated and independently trained models. As shown in Figure 3(a), AnaFP achieves the most favorable ROC curve that approaches the top left corner, while the curves of baseline methods lie relatively closer to the diagonal line, indicating weaker discriminative capability. This result further confirms AnaFP's superior performance in distinguishing pirated and independently trained models. On the other hand, Figure 3(b) complements this analysis by showing the distribution of fingerprint matching rates for both model sets with boxplots. The matching rate reflects the proportion of fingerprints for which a model returns the expected label. A greater separation between the distributions for pirated and independently trained models indicates higher discriminative capability. As depicted in the figure, AnaFP exhibits sharply separated distributions, enabling reliable verification outcomes. In contrast, the baselines show overlapped distributions, indicating ambiguity in their verification decisions.

**The robustness to performance-preserving model modification attacks**. We evaluate AnaFP's robustness to performance-preserving model modifications using seven representative attacks: pruning, fine-tuning, KD, AT, and three composite attacks (N-finetune, P-finetune, and Prune-KD). Table 2 shows AUCs obtained by AnaFP and the six baselines under each attack with CNN models. Under the pruning attack, all methods achieve near-perfect performance (AUC ≥ 0.999). However, AnaFP consistently outperforms the baselines under the remaining attacks, including fine-tuning, KD, AT, N-finetune, P-finetune, and Prune-KD. Notably, AnaFP achieves mean AUCs of 0.979, 0.983, 0.978, and 0.989 for fine-tuning, AT, N-finetune, and P-finetune, respectively, showing strong resistance to these attacks. Although KD and Prune-KD present a unique challenge by distilling knowledge without retaining internal structures and weights, AnaFP still maintains a leading AUC of 0.756 and 0.689, surpassing all baselines. These results collectively demonstrate AnaFP's robustness against a wide range of model modification attacks.

## 5.2 Sensitivity to Design Choices

### 5.2.1 The Impact of Surrogate Model Pool

To evaluate the robustness of AnaFP under different surrogate pool configurations, we examine two key factors: *pool size* and *pool diversity*.

**Sensitivity to pool size.** We examine the impact of surrogate pool size on verification performance by varying the number of models in each pool, considering configurations with 2, 4, 6, 8, and 10 models. As shown by the line plot in Figure 4, the AUC stabilizes rapidly once the pool size exceeds 6 models. This finding indicates that AnaFP achieves stable and reliable verification performance with only a modest number of surrogate models, indicating robustness to pool size.

**Sensitivity to pool diversity.** We further analyze the effect of surrogate set diversity on verification performance by constructing pools with varying levels of model diversity. In the *low-diversity* setting, the pirated model pool consists solely of fine-tuned variants of the protected model, and the independent model pool contains independently trained models sharing the same architecture as the protected model. The *medium-diversity* set-

Table 3: Effect of the quantile thresholds.

| $q_{\mathrm{margin}}/q_{\mathrm{lip}}$ | 0.1/0.9 | 0.2/0.8 | 0.3/0.7 | 0.4/0.6 | 0.5/0.5 | 0.6/0.4 | 0.7/0.3 | 0.8/0.2 | 0.9/0.1 |
|---|---|---|---|---|---|---|---|---|---|
| Valid Fingerprints | 0 | 0 | 11 | 33 | 56 | 103 | 228 | 428 | 1158 |
| AUC | – | – | 0.848 | 0.906 | 0.957 | 0.959 | 0.951 | 0.938 | 0.896 |

ting expands the pirated pool diversity to include both fine-tuned and knowledge-distilled models, while the independent pool comprises models with two distinct architectures. In the *high-diversity* setting, we further incorporate noise-finetuned pirated variants into the pirated pool and independently trained models with three different architectures in the independent pool. As shown in the bar plot of Figure 4, increasing diversity generally improves AUC. However, the performance gain from medium and high diversity is marginal, indicating diminishing returns. Importantly, even in the low-diversity setting, AnaFP maintains a strong AUC of 0.912, demonstrating its robustness to variations in surrogate pool diversity.

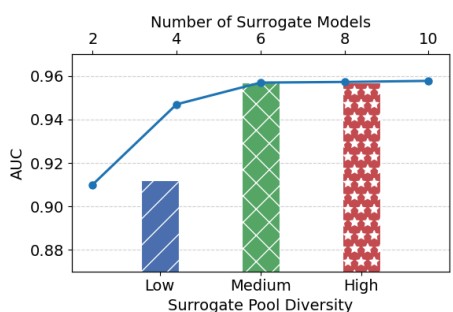

In summary, these results show that AnaFP is insensitive to the specific configurations of surrogate pools. Reliable ownership verification can be achieved using a modest number of surrogate models and without requiring extensive diversity, thereby underscoring AnaFP's practicality and generalizability.

Figure 4: Line plot (top axis): effect of surrogate pool size; Bar plot (bottom axis): effect of surrogate pool diversity.

### 5.2.2 THE IMPACT OF THE QUANTILE THRESHOLD

To assess how the quantile thresholds affect verification performance, we perform a sensitivity study on $q_{\mathrm{margin}}$ and $q_{\mathrm{lip}}$ while fixing $q_{\mathrm{eps}} = 1.0$, as its impact was found to be empirically negligible. The results are summarized in Table 3. Overly strict thresholds (e.g., $0.1/0.9$ and $0.2/0.8$) result in no valid fingerprints, making ownership verification impossible. Even with a slightly relaxed threshold like $0.3/0.7$, the small number of valid fingerprints remains limited (only 11), leading to a low AUC of 0.848. As the thresholds are moderately relaxed, the number of valid fingerprints increases, and verification performance quickly stabilizes. Notably, starting from the $0.5/0.5$ configuration, all subsequent threshold pairs yield AUC values exceeding 0.950, indicating that once a sufficiently large and reliable fingerprint set is established, the verification performance becomes robust to further threshold variations. However, excessive relaxation introduces a substantial number of low-quality fingerprints, ultimately degrading the discriminative capability (e.g., achieving the AUC of 0.896 at $0.9/0.1$). Overall, these findings indicate that AnaFP's verification performance is not highly sensitive to the precise choice of quantile thresholds. As long as the thresholds are moderately relaxed, a high-quality fingerprint pool can be obtained without extensive tuning.

### 5.3 ABLATION STUDY OF THE STRETCH FACTOR $\tau$

To assess the impact of the stretch factor $\tau$ on verification performance, we conduct an ablation study using five variants of AnaFP: A-lower (selecting $\tau$ from the feasible set $\mathcal{T}_{\mathrm{feas}}$ that is closest to the lower bound), A-upper (selecting $\tau$ from the feasible set $\mathcal{T}_{\mathrm{feas}}$ that is closest to the upper bound), C-fix (applying a fixed prediction margin across all fingerprints by optimizing $\tau$ to enforce the same difference between the probabilities of the first and second-highest predicted classes), C-lower (fixing $\tau$ to the lower bound $\tau_{\mathrm{lower}}$ to solely enforce robustness), and C-upper (setting $\tau$ to the upper bound $\tau_{\mathrm{upper}}$ to solely enforce uniqueness).

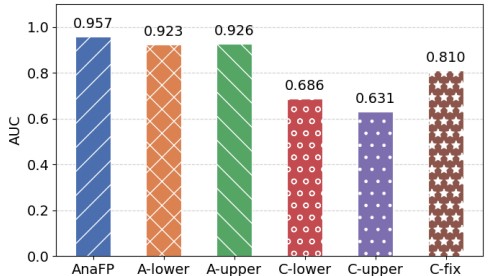

Figure 5: The AUCs achieved by AnaFP and three variants.

Figure 5 shows the AUC values achieved by AnaFP and its variants. Specifically, AnaFP consistently outperforms all five variants, validating the importance of searching for a feasible $\tau$ within its admissible interval. Among the variants, C-fix performs better than both C-lower and C-upper, as it

seeks a balance between robustness and uniqueness using a uniform margin. In contrast, C-lower and C-upper impose single constraints across all fingerprints—either robustness or uniqueness—without verifying whether both constraints are satisfied. Consequently, many infeasible $\tau$s are retained, degrading the overall discriminative capability. However, C-fix lacks the flexibility to adapt its margin to the local decision geometry of individual samples. On the other hand, AnaFP, A-lower, and A-upper discard infeasible $\tau$, thereby maintaining high verification reliability. While A-lower and A-upper exhibit slightly reduced performance, primarily due to the lower tolerance of their selected feasible $\tau$ to parameter-estimation errors, they still outperform C-fix, C-lower, and C-upper. These results underscore the importance of selecting $\tau$ that is within the admissible interval.

## 6 CONCLUSIONS

We study the problem of constructing effective fingerprints for adversarial-example-based model fingerprinting by ensuring simultaneous satisfaction of both robustness and uniqueness constraints. To this end, we propose AnaFP, an analytical fingerprinting scheme that mathematically formalizes the two constraints and theoretically derives an admissible interval, which is defined by the lower and upper bounds for the robustness and uniqueness constraints, for the stretch factor used in fingerprint construction. To enable practical fingerprint generation, AnaFP approximates the original model sets using two finite surrogate model pools and employs a quantile-based relaxation strategy to relax the derived bounds. Particularly, due to the circular dependency between the lower bound and the stretch factor, a grid search strategy is exploited to determine the most feasible stretch factor. Extensive experiments across diverse DNN architectures and datasets demonstrate that AnaFP achieves effective and reliable ownership verification, even under various model modification attacks, and consistently outperforms prior adversarial-example-based fingerprinting methods.

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

APPENDIX

## A   THEORETICAL DERIVATION OF THE LOWER AND UPPER BOUNDS

In Section 4.3, we introduced two theoretical constraints: a robustness constraint that defines a lower bound for $\tau$ and a uniqueness constraint that defines an upper bound for $\tau$. In this section, we present formal derivations of both bounds.

First, we provide Lemma A.1 about the lower bound for the robustness as follows.

**Lemma A.1** (Lower bound for robustness). *For the input of a fingerprint $\boldsymbol{x}^\star = \boldsymbol{x}^a + \tau\boldsymbol{\delta}^*$, the prediction of an arbitrary pirated model $P' \in \mathcal{V}_P$ is $P'(\boldsymbol{x}^\star) = y^\star$, provided that $\tau > 1 + \frac{2\,\epsilon_{logit}}{c_g\,\|\boldsymbol{\delta}^*\|_2}$.*

*Proof.* To guarantee $P'(\boldsymbol{x}^\star) = y^\star$, we have $s_{P',y^\star}(\boldsymbol{x}^\star) > s_{P',y}(\boldsymbol{x}^\star)$. For any pirated model $P'$, its logit shift $\epsilon_{\text{logit}}$ is bounded by $|s_{P',k}(\boldsymbol{x}^\star) - s_{P,k}(\boldsymbol{x}^\star)| \le \epsilon_{\text{logit}}, \forall k \in \mathcal{Y}$. Following this, for a label of the fingerprint $y^\star$ and the original label $y$, we have

$$s_{P',y^\star}(\boldsymbol{x}^\star) \ge s_{P,y^\star}(\boldsymbol{x}^\star) - \epsilon_{\text{logit}}, \qquad s_{P',y}(\boldsymbol{x}^\star) \le s_{P,y}(\boldsymbol{x}^\star) + \epsilon_{\text{logit}}.$$

Subtracting them yields $s_{P',y^\star}(\boldsymbol{x}^\star) - s_{P',y}(\boldsymbol{x}^\star) \ge g_P(\boldsymbol{x}^\star) - 2\epsilon_{\text{logit}}$. Hence, the robustness is guaranteed if the margin of the protected model satisfies

$$g_P(\boldsymbol{x}^\star) > 2\epsilon_{\text{logit}}. \tag{4}$$

We perform the first-order Taylor expansion at the boundary point $\boldsymbol{q} = \boldsymbol{x}^a + \boldsymbol{\delta}^*$, where $g_P(\boldsymbol{q}) = 0$. Letting $\boldsymbol{x}^\star = \boldsymbol{q} + (\tau - 1)\boldsymbol{\delta}^*$, we approximate $g_P(\boldsymbol{x}^\star) \approx (\tau - 1)\,\boldsymbol{\delta}^{*\top}\nabla g_P(\boldsymbol{q})$. Since $\boldsymbol{\delta}^*$ is the optimal direction that minimizes $\|\boldsymbol{\delta}\|_2$ while satisfying $g_P(\boldsymbol{x}^a + \boldsymbol{\delta}) = 0$, the KKT condition yields $\boldsymbol{\delta}^* = -\lambda\nabla g_P(\boldsymbol{q})$ for some $\lambda > 0$, implying colinearity. Therefore, we have $g_P(\boldsymbol{x}^\star) \approx (\tau - 1)\,c_g\|\boldsymbol{\delta}^*\|_2$. Plugging $g_P(\boldsymbol{x}^\star)$ into Equation equation 4, we find $(\tau - 1)\,c_g\|\boldsymbol{\delta}^*\|_2 > 2\epsilon_{\text{logit}}$, i.e., $\tau > 1 + \frac{2\epsilon_{\text{logit}}}{c_g\|\boldsymbol{\delta}^*\|_2}$, which concludes the proof. $\square$

Next, we state Lemma A.2 about the upper bound for the uniqueness as follows.

**Lemma A.2** (Upper bound for uniqueness). *For the input of a fingerprint $\boldsymbol{x}^\star = \boldsymbol{x}^a + \tau\boldsymbol{\delta}^*$, the original label $y$ of the anchor is preserved for $\forall I \in \mathcal{I}_P$, provided that $\tau < \frac{m_{\min}}{2\,L_{\text{uniq}}\,\|\boldsymbol{\delta}^*\|_2}$.*

*Proof.* We first fix an arbitrary independently trained model $I$ with Lipschitz constant $L_I \le L_{\text{uniq}}$. Let $\Delta = \tau\boldsymbol{\delta}^*$, and $k^\dagger = \arg\max_{k \neq y} s_{I,k}(\boldsymbol{x}^a)$ be the runner-up class. Then, the margin at the anchor is $g_I(\boldsymbol{x}^a) = s_{I,y}(\boldsymbol{x}^a) - s_{I,k^\dagger}(\boldsymbol{x}^a) \ge m_{\min}$. Now, setting $v = \arg\max_{k \neq y} s_{I,k}(\boldsymbol{x}^\star)$ and considering the margin at $\boldsymbol{x}^\star$, we have

$$g_I(\boldsymbol{x}^\star) = s_{I,y}(\boldsymbol{x}^\star) - s_{I,v}(\boldsymbol{x}^\star). \tag{5}$$

Because $L_{\text{uniq}}$ is an upper bound on the local Lipschitz constant across all independently trained models, we have $\left\|s_I(\boldsymbol{x}^\star) - s_I(\boldsymbol{x}^a)\right\|_2 \le L_{\text{uniq}}\|\Delta\|_2 = L_{\text{uniq}}\tau\|\boldsymbol{\delta}^*\|_2$, for $\forall I \in \mathcal{I}_P$. Each individual coordinate satisfies $|s_{I,k}(\boldsymbol{x}^\star) - s_{I,k}(\boldsymbol{x}^a)| \le \left\|s_I(\boldsymbol{x}^\star) - s_I(\boldsymbol{x}^a)\right\|_2$. Thus, we have $|s_{I,k}(\boldsymbol{x}^\star) - s_{I,k}(\boldsymbol{x}^a)| \le L_{\text{uniq}}\tau\|\boldsymbol{\delta}^*\|_2$. Following this equation, we have the following two inequalities:

$$s_{I,y}(\boldsymbol{x}^\star) \ge s_{I,y}(\boldsymbol{x}^a) - L_{\text{uniq}}\tau\|\boldsymbol{\delta}^*\|_2, \tag{6}$$
$$s_{I,v}(\boldsymbol{x}^\star) \le s_{I,v}(\boldsymbol{x}^a) + L_{\text{uniq}}\tau\|\boldsymbol{\delta}^*\|_2. \tag{7}$$

Subtracting equation 7 from equation 6 and inserting the result into equation 5, we have

$$\begin{aligned}
g_I(\boldsymbol{x}^\star) &\ge [s_{I,y}(\boldsymbol{x}^a) - s_{I,v}(\boldsymbol{x}^a)] - 2L_{\text{uniq}}\tau\|\boldsymbol{\delta}^*\|_2 \\
&\ge [s_{I,y}(\boldsymbol{x}^a) - s_{I,k^\dagger}(\boldsymbol{x}^a)] - 2L_{\text{uniq}}\tau\|\boldsymbol{\delta}^*\|_2 \quad (\text{since } s_{I,v}(x^a) \le s_{I,k^\dagger}(x^a)) \\
&= g_I(\boldsymbol{x}^a) - 2L_{\text{uniq}}\tau\|\boldsymbol{\delta}^*\|_2. \tag{8}
\end{aligned}$$

Preserving the original label $y$ at $\boldsymbol{x}^\star$ requires $g_I(\boldsymbol{x}^\star) > 0$, which implies $\tau < \frac{g_I(\boldsymbol{x}^a)}{2L_{\text{uniq}}\|\boldsymbol{\delta}^*\|_2}$. Since $g_I(\boldsymbol{x}^a) \ge m_{\min}$, we obtain $g_I(\boldsymbol{x}^\star) > 0$ for any $\tau < \frac{m_{\min}}{2L_{\text{uniq}}\|\boldsymbol{\delta}^*\|_2}$, which concludes the proof. $\square$

Following Lemmas A.1 and A.2, we establish Theorem A.1 about the robustness and uniqueness.

**Theorem A.1** (Admissible interval for $\tau$). *For $\forall P' \in \mathcal{V}_P$ and $\forall I \in \mathcal{I}_P$, any fingerprint $(\boldsymbol{x}^\star, y^\star)$ constructed with a feasible stretch factor $\tau^\star$ is guaranteed to be both robust and unique if $1 + \frac{2\,\epsilon_{logit}}{c_g\,\|\boldsymbol{\delta}^*\|_2} < \tau < \frac{m_{\min}}{2\,L_{uniq}\,\|\boldsymbol{\delta}^*\|_2}$.*

# B SUPPLEMENTAL EXPERIMENTAL RESULTS

## B.1 ROBUSTNESS TO MODEL MODIFICATION ATTACKS

Tables 4, 5, and 6 summarize the AUCs of AnaFP and the baseline methods under performance-preserving model modifications across CNN, MLP, and GNN architectures. For CNN models trained on CIFAR-100, AnaFP consistently achieves high AUCs ($\geq 0.930$) in five out of seven attacks, with the sole exception being KD and Prune-KD, where UAP and AKH marginally outperform AnaFP respectively. For MLP models, AnaFP attains near-perfect AUCs ($\geq 0.999$) on four attacks and maintains strong performance on AT with an AUC of 0.987; besides, AnaFP performs best under KD and Prune-KD. In contrast, for GNN models, AnaFP achieves the highest AUC across all six attacks, with AUCs, ranging from 0.736 to 0.971, significantly outperforming IPGuard and AKH. It is worth noting that UAP, MarginFinger, ADV-TRA, and GMFIP are not directly applicable to graph-structured data, as they assume a fixed input geometry. The consistently superior performance achieved by AnaFP across diverse model types highlights its strong generalization ability and robustness to modification attacks, which is particularly important for ownership verification in diverse real-world deployment scenarios.

### B.1.1 THE IMPACT OF ANCHOR SELECTION

We investigate the impact of confidence levels during anchor selection by performing experiments with low-confidence, mid-confidence, and high-confidence anchors on CNN(CIFAR-10) and MLP(MNIST). The three confidence levels are defined by the range of logit margins $g_P(x^a)$. Experimental results illustrate that low-confidence anchors ($0 \leq g_P(x^a) < 2.5$) yield 0 valid fingerprints for both cases, mid-confidence anchors ($2.5 \leq g_P(x^a) < 5.0$) produces 3 valid fingerprints for CNN and 2 for MLP with the achieved AUCs of 0.791 and 0.755, respectively, and high-confidence anchors ($g_P(x^a) \geq 5.0$) produce 57 valid fingerprints for CNN and 85 for MLP with much higher AUC (i.e., 0.958 for CNN and 0.961 for MLP). This confirms that low-confidence anchors fail to produce a sufficient number of valid fingerprints and thus degrade the performance.

### B.1.2 ROBUSTNESS TO THE KNOWLEDGE DISTILLATION ATTACK

In addition to the main metric AUC, which reflects the overall detection capability across all threshold settings, we will also assess TPR, FPR, TNR, and FNR for the KD attack. The experimental results are summarized in the Table 7. From this table, we can observe that AnaFP achieves the strongest separation between pirated models under the KD attack and independently trained models, reaching a high true positive rate (TPR=0.80) while keeping the false positive rate relatively low (FPR=0.22). ADV-TRA and UAP perform worse than AnaFP but better than other baselines, showing a moderate-level performance (TPR=0.73, FPR=0.30 for UAP, and TPR=0.75, FPR=0.27 for ADV-TRA). In contrast, AKH, IPGuard, GMFIP, and MarginFinger are relatively less capable of distinguishing

Table 4: AUCs under various performance-preserving model modifications on CNN models trained on CIFAR100.

| Method | Pruning | Finetuning | KD | AT | N-finetune | P-finetune | Prune-KD |
|---|---|---|---|---|---|---|---|
| AnaFP (ours) | **0.963 ± 0.002** | **0.954 ± 0.004** | 0.596 ± 0.009 | **0.930 ± 0.007** | **0.957 ± 0.005** | **0.956 ± 0.004** | 0.574 ± 0.013 |
| UAP | 0.897 ± 0.015 | 0.852 ± 0.022 | **0.598 ± 0.018** | 0.802 ± 0.027 | 0.852 ± 0.022 | 0.859 ± 0.021 | 0.583 ± 0.019 |
| IPGuard | 0.845 ± 0.086 | 0.811 ± 0.092 | 0.461 ± 0.096 | 0.721 ± 0.080 | 0.763 ± 0.088 | 0.750 ± 0.098 | 0.496 ± 0.078 |
| MarginFinger | 0.737 ± 0.076 | 0.667 ± 0.092 | 0.519 ± 0.026 | 0.679 ± 0.092 | 0.653 ± 0.089 | 0.553 ± 0.058 | 0.488 ± 0.026 |
| AKH | 0.915 ± 0.007 | 0.838 ± 0.032 | 0.592 ± 0.034 | 0.762 ± 0.076 | 0.829 ± 0.040 | 0.878 ± 0.044 | **0.601 ± 0.041** |
| ADV-TRA | 0.953 ± 0.007 | 0.904 ± 0.024 | 0.561 ± 0.011 | 0.886 ± 0.035 | 0.901 ± 0.017 | 0.898 ± 0.031 | 0.544 ± 0.030 |
| GMFIP | 0.885 ± 0.048 | 0.794 ± 0.076 | 0.548 ± 0.065 | 0.721 ± 0.078 | 0.877 ± 0.090 | 0.902 ± 0.064 | 0.516 ± 0.072 |

Table 5: AUCs under various performance-preserving model modifications on MLP models.

| Method | Pruning | Finetuning | KD | AT | N-finetune | P-finetune | Prune-KD |
|---|---|---|---|---|---|---|---|
| AnaFP (ours) | **1.000 ± 0.000** | **1.000 ± 0.000** | **0.792 ± 0.002** | **0.987 ± 0.005** | **1.000 ± 0.000** | **0.999 ± 0.000** | **0.788 ± 0.006** |
| UAP | 0.981 ± 0.001 | 0.969 ± 0.006 | 0.635 ± 0.002 | 0.931 ± 0.010 | 0.968 ± 0.007 | 0.976 ± 0.003 | 0.610 ± 0.005 |
| IPGuard | 0.924 ± 0.042 | 0.983 ± 0.007 | 0.535 ± 0.036 | 0.902 ± 0.029 | 0.938 ± 0.008 | 0.922 ± 0.037 | 0.517 ± 0.029 |
| MarginFinger | 0.708 ± 0.130 | 0.750 ± 0.145 | 0.572 ± 0.041 | 0.762 ± 0.149 | 0.759 ± 0.146 | 0.586 ± 0.090 | 0.570 ± 0.034 |
| AKH | 0.849 ± 0.017 | 0.870 ± 0.023 | 0.765 ± 0.008 | 0.897 ± 0.026 | 0.864 ± 0.025 | 0.777 ± 0.009 | 0.759 ± 0.011 |
| ADV-TRA | 0.971 ± 0.010 | 0.964 ± 0.007 | 0.596 ± 0.011 | 0.907 ± 0.005 | 0.922 ± 0.004 | 0.953 ± 0.001 | 0.602 ± 0.005 |
| GMFIP | 0.978 ± 0.021 | 0.971 ± 0.006 | 0.696 ± 0.042 | 0.914 ± 0.038 | 0.945 ± 0.029 | 0.873 ± 0.017 | 0.651 ± 0.054 |

Table 6: AUCs under various performance-preserving model modifications on GNN models.

| Method | Pruning | Finetuning | KD | AT | N-finetune | P-finetune | Prune-KD |
|---|---|---|---|---|---|---|---|
| AnaFP (ours) | **0.933 ± 0.002** | **0.971 ± 0.007** | **0.806 ± 0.002** | **0.970 ± 0.012** | **0.962 ± 0.013** | **0.913 ± 0.006** | **0.736 ± 0.015** |
| IPGuard | 0.806 ± 0.051 | 0.668 ± 0.053 | 0.529 ± 0.330 | 0.612 ± 0.122 | 0669 ± 0.027 | 0.551 ± 0.074 | 0.571 ± 0.174 |
| AKH | 0.859 ± 0.011 | 0.831 ± 0.043 | 0.790 ± 0.021 | 0.869 ± 0.045 | 0.838 ± 0.041 | 0.849 ± 0.038 | 0.719 ± 0.036 |

pirated models under the KD attack from independently trained models, reflected by their high FPR (0.36-0.47) and only moderately high TPR (0.57-0.67).

## B.2 EFFICIENCY ANALYSIS

We experimentally compare the computing time and peak GPU memory consumed by AnaFP and baselines. The experimental results are summarized in Table 8. Specifically, AKH and IPGuard are the most lightweight methods: AKH operates purely through forward inference, and IPGuard performs small-batch gradient updates without minimizing perturbation magnitude, resulting in runtimes of around one minute and memory usage below 3 GB. UAP is the most expensive method, as it requires multiple full passes over the training data to optimize a universal perturbation, leading to high computing time and memory usage. Both MarginFinger and GMFIP achieve moderate cost by training a generator model to synthesize boundary-adjacent samples. ADV-TRA generates adversarial trajectories as fingerprints, which causes high time cost due to the sequential generation process. Compared to these baselines, AnaFP consumes high memory but achieves balanced runtime. More specifically, AnaFP costs the most GPU memory—up to 40 GB on ViT-S/16, as it processes a large batch of anchors in parallel during the adversarial optimization step. This parallel design reduces the total number of optimization iterations, thus reducing the overall runtime. Despite the high memory usage, AnaFP can still run on a single A100 GPU, making the cost manageable in practice.

Moreover, we profiled the runtime of the three steps in our proposed fingerprinting method: Step 1 (anchor selection), Step 2 (adversarial perturbation computation), and Step 3 (searching stretch factors). We also examine the impact of the model sizes (ResNet18 and ViT-S/16) and the number of surrogate models used. The results are presented in the Table 9. From the table, we can see that the runtime cost is acceptable. Specifically, first, Step 2 dominates the total runtime, requiring 1000–3800 seconds depending on the protected model size and surrogate pool size, whereas the time cost of Step 1 is negligible, and Step 3 contributes a moderate amount (1000–2100 seconds). Second, the time cost scales with model size: fingerprinting ViT-S/16 requires significantly more time cost than fingerprinting ResNet-18, because a larger model incurs more computations during the fingerprint generation process. Third, increasing the surrogate pool from 4 to 8 models increases the runtime of Step 3, where grid search accounts for the major time cost. From the above discussion, we can see AnaFP's runtime is moderate.

We also measured GPU memory usage during Step 2 and Step 3, the experimental results are summarized in Table 10. We can observe that first, Step 2 requires the highest memory because it performs adversarial optimization, consuming 13 GB for ResNet-18 and about 40 GB for ViT-S/16, whereas Step 3 requires 1–3 GB. Second, memory usage grows with model size: ViT-S/16 on Tiny-ImageNet has roughly three times the memory requirements of ResNet-18 due to a larger model size. Third, increasing the surrogate pool size from 4 to 8 models increases memory usage slightly.

**Discussion on Potential Speedups.** The computational efficiency of AnaFP can be further improved through the following practices. First, regarding surrogate pools, independent surrogate models can

Table 7: Detection performance under the KD attack for CNN/CIFAR10.

| Method | TPR | FPR | TNR | FNR |
|---|---|---|---|---|
| AnaFP (ours) | **0.80** | **0.22** | 0.78 | 0.20 |
| UAP | 0.73 | 0.30 | 0.70 | 0.27 |
| IPGuard | 0.58 | 0.46 | 0.54 | 0.42 |
| MarginFinger | 0.57 | 0.47 | 0.53 | 0.43 |
| AKH | 0.63 | 0.40 | 0.60 | 0.37 |
| ADV-TRA | 0.75 | 0.27 | 0.73 | 0.25 |
| GMFIP | 0.67 | 0.36 | 0.64 | 0.33 |

Table 8: Time cost and peak GPU memory consumption of different fingerprinting methods under two representative protected models.

| | ResNet-18 | | ViT-S/16 | |
|---|---|---|---|---|
| | Time | Memory | Time | Memory |
| AnaFP (ours) | 33 m 8 s | 13 GB | 1 h 26 m | 40 GB |
| UAP | 4 h 55 m | 4 GB | 8 h 10 m | 10 GB |
| IPGuard | 1 m 6 s | 1 GB | 1 m 41 s | 3 GB |
| MarginFinger | 51 m 18 s | 4 GB | 1 h 14 m | 10 GB |
| AKH | 20 s | 1 GB | 1 m 10 s | 3 GB |
| ADV-TRA | 58 m 01 s | 4 GB | 1 h 29 m | 10 GB |
| GMFIP | 1 h 15 m | 4 GB | 1 h 40 m | 10 GB |

be drawn directly from publicly available models, eliminating the need for additional training, while generating pirated surrogate models via fine-tuning or pruning incurs very low cost. Even performing knowledge distillation, which is relatively more expensive, remains cheaper than training models from scratch. Second, the $C\&W$ optimization used in Step 2 (computing minimal decision-altering perturbations) can be replaced with more efficient alternatives, such as PGD (Madry et al., 2018) and DeepFool (Moosavi-Dezfooli et al., 2015), which can reduce computations. Third, a coarse-to-fine search strategy (Rubinstein, 1999) can be applied to further reduce the computational cost.

## C    IMPLEMENTATION DETAILS

### C.1    MODEL ARCHITECTURES AND TRAINING SETTINGS

We summarize the architectures and training configurations of the protected model, the attacked models, and the independently trained models used for each task in the following.

#### C.1.1    PROTECTED MODEL

**CNN (CIFAR-10 and CIFAR-100).** The protected model is a ResNet-18 trained from scratch using SGD with momentum 0.9, weight decay 5e-4, learning rate 0.1, cosine annealing learning rate scheduling, and Xavier initialization. Training is conducted for 600 epochs with a batch size of 128.

**MLP (MNIST).** The protected model is a ResMLP trained from scratch using SGD with momentum 0.9, weight decay 5e-4, a learning rate of 0.01, cosine annealing learning rate scheduling, and Kaiming initialization (default Pytorch setting). Training is conducted for 40 epochs with a batch size of 64.

**GNN (PROTEINS).** The protected model is a Graph Attention Network (GAT) trained using the Adam optimizer with a learning rate of 0.01 and Xavier initialization (default Pytorch setting). The default PyTorch Adam settings are used: $\beta_1 = 0.9$ and $\beta_2 = 0.999$.

#### C.1.2    INDEPENDENTLY TRAINED MODELS

**CNN (CIFAR-10 and CIFAR-100).** The independent model set consists of ten diverse architectures: ResNet-18, ResNet-50, ResNet-101, WideResNet-50, MobileNetV2, MobileNetV3-Large, EfficientNet-B2, EfficientNet-B4, DenseNet-121, and DenseNet-169. Each model is trained from

Table 9: Time cost for the three fingerprint-generation steps: Step 1 (anchor selection), Step 2 (adversarial perturbation generation), and Step 3 ($\tau$ search).

| Protected Model | Surrogate pool size | Step 1 | Step 2 | Step 3 |
|---|---|---|---|---|
| ViT-S/16 | 4 | 63s | 3856s | 1276s |
| ResNet18 | 4 | 8s | 1004s | 976s |
| ViT-S/16 | 8 | 61s | 3855s | 2121s |
| ResNet18 | 8 | 8s | 997s | 1419s |

Table 10: Peak memory consumption for Step 2 (adversarial perturbation generation) and Step 3 (grid search).

| Protected Model | Surrogates | Step 2 | Step 3 |
|---|---|---|---|
| ViT-S/16 | 4 | 40G | 3G |
| ResNet18 | 4 | 13G | 1G |
| ViT-S/16 | 8 | 41G | 3.2G |
| ResNet18 | 8 | 13G | 1.1G |

scratch with different random seeds under the same optimizer, scheduler, and initialization scheme as the protected model.

**MLP (MNIST).** The independent model set includes seven multilayer perceptron architectures: WideDeep, ResMLP, FTMLP, SNNMLP, NODE, TabNet, and TabTransformer. Each model is trained from scratch under the same optimizer, learning rate, scheduler, and number of epochs as the protected model, but with different random seeds and architecture-specific parameters where applicable.

**GNN (PROTEINS).** The independent model set includes seven graph neural network architectures: GCN, GIN, GraphSAGE, GAT, GATv2, SGC, and APPNP. Each model is trained from scratch using the same optimizer, learning rate, and number of epochs as the protected model, but with different random seeds and architecture-specific configurations.

### C.1.3 PIRATED MODELS

We construct pirated models across all tasks using seven types of performance-preserving attacks applied to the protected model with the following settings.

- **Fine-tuning**: retraining the protected model for a specified number of epochs.

- **Adversarial Training (AT)**: retraining the protected model for a specified number of epochs using a mix of normal data samples and adversarial samples.

- **Pruning**: applying unstructured global pruning with sparsity levels ranging from 10% to 90% in increments of 10%, without retraining.

- **P-Finetune**: applying pruning at sparsity levels of 30%, 60%, and 90%, followed by fine-tuning.

- **N-Finetune**: perturbing each trainable parameter tensor with Gaussian noise scaled by its standard deviation, i.e., $param += 0.09 \times std(param) \times \mathcal{N}(0, 1)$, followed by fine-tuning.

- **Knowledge Distillation (KD)**: training a student model of the same or different architecture with the protected model using the KL divergence between the outputs of the protected (teacher) model and the student model, temperature $T$ is set to 1.

- **Prune-KD**: performing pruning to the protected model, followed by knowledge distillation (KD) attack.

All attacks are applied consistently across tasks, with differences only in optimizer settings, learning rates, and the number of fine-tuning epochs. For **CNN (CIFAR-10)**, we use SGD with a learning rate of 0.01 and a cosine annealing scheduler for 200 epochs (600 epochs for KD). For **MLP (MNIST)**, the optimizer is SGD with a learning rate of 0.001, also using cosine annealing over 30 epochs (30 epochs for KD). For **GNN (PROTEINS)**, we use the Adam optimizer with a learning rate of 0.001 for 60 epochs (100 epochs for KD).

### C.1.4 DATA PREPROCESSING

All datasets are normalized prior to training the DNN models, in line with standard preprocessing practice. For the MNIST dataset, each $28 \times 28$ grayscale image is flattened into a 784-dimensional vector to serve as the input to the MLP model. For CIFAR-10, CIFAR-100, and PROTEINS, no additional preprocessing is applied beyond normalization.

## C.2 THE IMPLEMENTATION DETAILS OF ANAFP

### C.2.1 ANCHOR SELECTION AND ADVERSARIAL PERTURBATION

To identify anchors, we use a logit margin threshold of $m_{\text{anchor}} = 1.0$ for GNN on PROTEINS, and $m_{\text{anchor}} = 8.0$ for all other tasks. To compute the minimal perturbation that can alter prediction results (cf. Step 2 of AnaFP), we adopt the C&W-$\ell_2$ attack. The attack is parameterized with a confidence margin $kappa = 0$, the number of optimization steps $steps = 3000$, and the optimization learning rate $lr = 0.01$. We set the regularization constant $c$ per model–dataset pair, since the difficulty of finding decision-altering adversarial perturbations varies across models and dataset; for example, we use $c = 10^{-4}$ for CNNs on CIFAR-10 and CIFAR-100, while using $c = 0.9$ for MLPs on MNIST and $c = 10^{-2}$ for GNNs on PROTEINS. In practice, we recommend tuning $c$ based on the number of successfully obtained boundary points, increasing $c$ if too few boundary points are found.

### C.2.2 SURROGATE MODEL POOLS CONSTRUCTION

To enable estimation of theoretical bounds in Step 3 of AnaFP, we construct two surrogate model pools for each task: a pirated model pool consisting of six models obtained via two simulated performance-preserving modifications of the protected model (fine-tuning and knowledge distillation), and an independent model pool consisting of six independently trained models trained from scratch with different architectures and random seeds. All surrogate models and test models are trained with different random seeds and architectures, ensuring no surrogate models overlap with any test models.

Specifically, for the CNN on CIFAR-10 and CIFAR-100, pirated models include fine-tuned and knowledge-distilled models using ResNet-18 and DenseNet-121. The architectures of independently trained models include ResNet-18 and DenseNet-169, with three random seeds per architecture. For the MLP on MNIST, pirated models include fine-tuned ones, along with knowledge-distilled models using FT-MLP and Wide&Deep. The architectures of independently trained models consist of NODE and ResMLP, each instantiated with three seeds. For the GNN task on PROTEINS, pirated models include fine-tuned ones, as well as knowledge-distilled models with GAT and GATv2. The architectures of independently trained models consist of APPNP and GAT, each trained with three random seeds.

In principle, the surrogate pool should be as diverse as possible; however, under a constrained budget, we prioritize challenging surrogates—e.g., knowledge-distilled variants—in the pirated pool, and include independently trained models that share the same architecture with the protected model in the independent pool, since such models are behaviorally closer to the protected model and therefore yield a stricter and more informative bound.

### C.2.3 PARAMETER ESTIMATION

The three parameters $m_{\text{min}}$, $L_{\text{uniq}}$, and $\epsilon_{\text{logit}}$ are estimated using the surrogate model pools described in Appendix C.2.2. For each anchor, the lower bound $m_{\text{min}}$ of the logit margin is estimated by computing the logit margin for each surrogate independently trained model $I$, and aggregating using the $q_{\text{margin}}$-quantile. The local Lipschitz constant $L_{\text{uniq}}$ is estimated for each independently trained model based on the ratio of logit difference to input norm between the anchor input $x^a$ and its boundary point $x^a + \delta^*$, and aggregated via the $q_{\text{lip}}$-quantile. The bound $\epsilon_{\text{logit}}$ of the logit shift bound is computed as the maximum per-class logit difference between the protected model and the pirated ones at the perturbed input $x^\star$, and relaxed using $q_{\text{eps}}$-quantile. The quantile configurations are as follows. For the CNN model evaluated on CIFAR-10 and MLP on MNIST, $q_{\text{margin}}$, $q_{\text{lip}}$, and $q_{\text{eps}}$ are set to be 0.5, 0.5, and 1.0, respectively. For GNN on PROTEINS, we set $q_{\text{margin}}$, $q_{\text{lip}}$, and $q_{\text{eps}}$ to be 0.4, 0.6, and 1.0, respectively. For CNN on CIFAR-100, we set $q_{\text{margin}}$, $q_{\text{lip}}$, and $q_{\text{eps}}$ are set to be 0.3, 0.7, and 1.0, respectively. In practice, though the optimal values often vary across different scenarios,

selecting the thresholds requires only coarse adjustment, typically by gradually relaxing them from the conservative end until a reasonable number of valid fingerprints is found (e.g., 20–100), which is corroborated by the experimental results in Section 5.2.2.

### C.2.4 GRID SEARCH

The grid search over the interval $(1, \tau_{\text{upper}}]$ is implemented by uniformly sampling a finite number of stretch factor value candidates. Specifically, we sample $N_{\text{grid}}$ points within this interval, where $N_{\text{grid}} = 500$ is used by default in all our experiments. Increasing $N_{\text{grid}}$ enables a finer-grained search over $(1, \tau_{\text{upper}}]$ and may yield better performance in principle, but it also increases the computational cost proportionally.

## D SCOPE AND APPLICABILITY

AnaFP is fundamentally task-dependent, as it constructs and verifies fingerprints based on class decision boundaries. As a result, like most existing fingerprinting methods, it is naturally designed for classification tasks. This dependence limits its applicability in scenarios where well-defined class boundaries do not exist, such as regression problems (which produce continuous outputs without discrete decision regions) and many self-supervised / unsupervised models whose objectives are not label-based (e.g., representation learning without a classifier head). Extending AnaFP to these settings would likely require redefining the fingerprint signal around alternative structural properties, such as similarity geometry in embedding space or task-specific surrogate decision functions. We leave this extension to future work.

## E THE COMPUTATION PLATFORM

All experiments are conducted on a high-performance server equipped with an NVIDIA A100 GPU with 40 GB of memory and an Intel Xeon Gold 6246 CPU running at 3.30 GHz. The software environment includes Python 3.9.21, PyTorch 2.5.1, and CUDA 12.5.

## F RELATED WORK

Model fingerprinting has become a key paradigm for protecting intellectual property (IP) of DNN models (Cao et al., 2021; Guan et al., 2022; Wang et al., 2021a). In contrast to watermarking that embeds additional functionalities into model parameters, which introduces security issues to the model (Wang et al., 2021a; Xu et al., 2024; Li et al., 2024; Choi et al., 2025), fingerprinting provides a non-intrusive alternative that protects a model's ownership by extracting its intrinsic decision behavior without making any modifications to it. The majority of existing methods consider a practical black-box setting where ownership verification is conducted by querying the suspect model with fingerprints and comparing the similarity between the outputs of the protected model and the suspect model (Cao et al., 2021; Xu et al., 2024; Zhao et al., 2024a; You et al., 2024; Ren et al., 2023; Yin et al., 2022; Yang & Lai, 2023; Peng et al., 2022; Liu & Zhong, 2024; Godinot et al., 2025).

Among these methods, many works leverage adversarial examples as fingerprints to induce model-specific outputs for the protected models (Zhao et al., 2020; Yin et al., 2022; Peng et al., 2022; Yang & Lai, 2023). To improve robustness against ownership obfuscation techniques, several studies create variants for the protected model to optimize fingerprints (Wang et al., 2021a;b). To further enhance the detection capability, Lukas et al. (2021); Li et al. (2021); Yang et al. (2022); Ren et al. (2023) train fingerprints to yield consistent outputs across pirated variants of the protected model while remaining distinguishable from independently trained models. Besides, some methods extract robust, task-relevant features such as universal perturbations or adversarial trajectories to construct stronger identifiers Peng et al. (2022); Xu et al. (2024). Recently, Godinot et al. (2025) leverages the training samples that have been misclassified by the protected model as fingerprints, which demonstrate stronger discriminative power than normal adversarial examples. Tang et al. (2025) proposes a robust fingerprinting scheme that embeds ownership information in the frequency domain. Zhang et al. (2025) identifies the most informative fingerprints from the set to achieve higher verification performance under limited query budgets.

For model fingerprinting, an open question is how to craft fingerprints that can satisfy the properties of robustness (robust to model modification attacks) and uniqueness (not falsely attribute independently trained models as pirated) simultaneously. Several previous works provided their answers, i.e., empirically controlling the placement of fingerprints relative to the decision boundaries (Cao et al., 2021; Liu & Zhong, 2024). However, these approaches rely on empirically tuned distance without any theoretical guidance, which can lead to unstable fingerprint behavior and inconsistent verification performance. In this paper, we propose an analytical fingerprinting approach that crafts adversarial example-based fingerprints under the guidance of formalized properties of robustness and uniqueness.

## G  SOCIETAL IMPACTS

Our work proposes a fingerprinting framework for verifying the ownership of deep neural network (DNN) models, aiming to protect the intellectual property (IP) of model creators. By enabling reliable and robust ownership verification, our method contributes positively to the development of secure and trustworthy AI ecosystems. This can help mitigate unauthorized usage, model stealing, and commercial misuse, encouraging responsible AI deployment and fair attribution in both academia and industry.

