# OpenReview forum: "Fingerprinting Deep Neural Networks for Ownership Protection: An Analytical Approach"
_ICLR.cc/2026/Conference — ICLR 2026 Poster_

### Official Review · Reviewer_UJXn · 2025-10-22

**Soundness:** 3
**Presentation:** 4
**Contribution:** 3
**Rating:** 8
**Confidence:** 3

**Summary:**

This paper proposes an adversarial-example-based fingerprinting approach for Deep Neural Networks (DNNs). It investigates the fingerprint-to-boundary distance, which is a fundamental challenge in current DNN fingerprinting methods. By introducing a tunable stretch factor and determine the lower and upper bounds of this factor, the authors mathematically define an admissible interval within which the fingerprint-to-boundary distance must lie.

**Strengths:**

The paper is well-written and clearly introduces the challenges associated with current DNN fingerprinting methods as well as the proposed solution.


The paper offers an effective solution for quantitatively characterizing model boundaries for models that function similarly or identically but originate from different sources.

**Weaknesses:**

The effectiveness of the proposed method heavily depends on the prior knowledge provided by the 120 pirated-independent model pairs in the model pool. To protect a model, it is necessary to construct a model pool, and techniques such as Knowledge Distillation (KD) and Adversarial Training (AT) introduce additional overheads that may limit practical applications.

**Questions:**

The authors claim that selecting high-confidence samples ensures strong uniqueness. However, logically, the closer the fingerprint is to the decision boundary, the more likely it is to ensure uniqueness. In generating adversarial fingerprint samples, a smaller update step size should, in theory, bring the fingerprint closer to the decision boundary. Selecting high-confidence samples as anchors could potentially increase the number of iterations, but this may not necessarily reduce the distance of the generated adversarial fingerprint sample from the decision boundary. If time permits, I would appreciate it if the authors could provide experimental results using mid-confidence or low-confidence examples as anchors.

---

> ### Author Response · Authors · 2025-11-21
>
> ## Concern 1:
>
> The effectiveness of the proposed method heavily depends on the prior knowledge provided by the 120 pirated-independent model pairs in the model pool. To protect a model, it is necessary to construct a model pool, and techniques such as Knowledge Distillation (KD) and Adversarial Training (AT) introduce additional overheads that may limit practical applications.
>
> **Response to concern 1:**
> We would like to clarify that in our paper, we have two types of model sets: one is the surrogate piracy and independent pools for estimating theoretical bounds, and the other is the pirated and independent model sets used for experimentally evaluating AnaFP's performance. The models in these model pools/sets are independently trained with no overlap, ensuring that each model is entirely unseen by the others.
>
> The 120 pirated-independent model pairs mentioned by the reviewer are used exclusively for experimentally evaluating AnaFP's performance, not for estimating theoretical bounds in our method. In fact, AnaFP requires only two small-size surrogate model pools. As shown in Sec. 5.2.1, these surrogate pools can be extremely small in practice: using only 4–6 models already achieves nearly optimal AUC, and even 2 models yield AUC $>$ 0.9. Moreover, Sec. 5.2.1 further shows that the surrogate pirated pool does not require broad attack diversity. Our experiments confirm that incorporating only two simple attacks (fine-tuning and KD) in the surrogate piracy pool is sufficient to attain strong generation across a testing pool that includes six diverse attack types. These findings highlight the practicality and generalizability of AnaFP in practical settings.
>
> ---
>
> ## Question 1:
>
> The authors claim that selecting high-confidence samples ensures strong uniqueness. However, logically, the closer the fingerprint is to the decision boundary, the more likely it is to ensure uniqueness. In generating adversarial fingerprint samples, a smaller update step size should, in theory, bring the fingerprint closer to the decision boundary. Selecting high-confidence samples as anchors could potentially increase the number of iterations, but this may not necessarily reduce the distance of the generated adversarial fingerprint sample from the decision boundary. If time permits, I would appreciate it if the authors could provide experimental results using mid-confidence or low-confidence examples as anchors.
>
> **Response to Question 1:**
> Uniqueness is not determined by the anchor’s proximity to the protected model’s boundary, but by the anchor’s margin under independently trained models. High-confidence anchors maximize this margin ($m_{min}$), which directly enlarges the admissible interval for the stretch factor $\tau$ by enlarging $\tau_{upper}$ in Eq. (3). Low-confidence anchors can reduce the value of $\tau_{upper}$, narrowing the admissible interval and decreasing the probability of finding a feasible $\tau$. Thus, the number of valid fingerprints found in this process is small, which significantly degrades the fingerprinting performance. We conducted experiments with low-confidence, mid-confidence, and high-confidence anchors. Experimental results illustrate that low-confidence anchors (0 $\leq g_P(x^a) <$ 2.5) yield 0 valid fingerprints for both CIFAR-10 and MNIST, mid-confidence anchors ($2.5 \leq g_P(x^a) < 5.0$) produce 3/2 valid fingerprints and achieve AUCs of 0.791/0.755 respectively, and high-confidence anchors ($g_P(x^a) \geq $ 5.0) produce 57/85 valid fingerprints and achieve much higher AUC (0.958/0.961). This confirms that low-confidence anchors fail to produce a sufficient number of valid fingerprints and thus degrade the performance.

---

> > ### Comment · Reviewer_UJXn · 2025-11-23
> >
> > Thansk for the author's responses. I keep my positive score.

---

> > > ### Author Response · Authors · 2025-11-24
> > >
> > > Thank you so much! We appreciate your constructive feedback and are glad that our responses addressed your concerns.

---

### Official Review · Reviewer_ZKy4 · 2025-10-28

**Soundness:** 2
**Presentation:** 3
**Contribution:** 2
**Rating:** 4
**Confidence:** 3

**Summary:**

This paper introduces AnaFP, an analytical framework for generating adversarial-example-based fingerprints for deep neural network model ownership verification. The key contribution is the mathematical formalization and unified treatment of both robustness and uniqueness constraints, enabling derivation of a theoretically principled interval for positioning fingerprints relative to the decision boundary. AnaFP employs surrogate model pools and a quantile relaxation strategy to make implementation practical and robust to variations in model modification and diversity.

**Strengths:**

- The paper rigorously formalizes the requirements for fingerprint robustness and uniqueness, deriving both lower and upper bounds on the stretch factor that controls fingerprint placement.

- AnaFP is demonstrated on a diverse set of DNN models, indicating broad applicability. Unlike some baselines (UAP, MarginFinger), AnaFP naturally extends to non-Euclidean domains.

- The paper provides well-designed ablations on surrogate pool size/diversity, quantile relaxation parameters, and the selection of the stretch factor τ.

- The methodology is well structured and enhanced with explanatory figures.

**Weaknesses:**

- While the authors thoroughly test robustness to model modifications and evaluate discriminability between pirated and independent models, there is no assessment of scenarios with "unknown" models or more ambiguous conditions (e.g., surrogate attacks specifically designed to evade fingerprints; adaptive adversaries targeting decision geometry). Similarly, detailed analysis of the potential for false positives in more open-world settings is missing.

- AnaFP's practicality depends on the construction of surrogate pirated and independent pools. While sensitivity to pool size/diversity is empirically analyzed, the method still assumes access to representative pools, which may not always be feasible—especially in dynamic, heterogeneous model deployment ecosystems. There is little discussion of how errors in pool selection could impact the upper/lower bound estimation and thus fingerprint viability.

- Although quantile-based relaxation is well motivated, the chosen quantiles are somewhat arbitrary and task dependent. The method remains sensitive to settings that are not fully justified theoretically.

- The fingerprint creation protocol may incur significant computational costs for large-scale or production systems. There is no empirical profiling of runtime or scalability as the number of fingerprints, surrogates, or model size increases.

**Questions:**

- How would AnaFP fare against adaptive attackers who are aware of the fingerprint generation protocol and attempt to evade by specifically distorting or randomizing localized decision boundaries? Are there (perhaps adversarially optimized) model modifications that could reduce the efficacy of the derived interval without violating standard performance constraints?

- Can the authors clarify or provide more guidance on choosing quantile thresholds for the relaxation strategy (see Table 3), beyond the trial-and-error or “reasonable number of fingerprints” heuristic? Any analytical or automated procedure to further reduce parameter sensitivity?

- The use of grid search with potentially many anchors and surrogate models may be computationally expensive, especially for large-scale or production systems. Could you provide more details on the runtime complexity and memory requirements for AnaFP in realistic settings?

- The evaluation focuses on known model modification attacks (pruning, fine-tuning, KD, AT). Have you considered or tested robustness against adaptive attackers who might specifically attempt to evade fingerprinting by perturbing near-anchor regions or strategically altering decision boundaries?

- While AnaFP shows strong results on CNNs, MLPs, and GNNs, are there any specific architectural choices, dataset characteristics, or domains where the approach might face limitations? For example, very large-scale models or models trained with self-supervised or unsupervised objectives?

---

> ### Author Response · Authors · 2025-11-21
>
> ## Concern 1:
>
> While the authors thoroughly test robustness to model modifications and evaluate discriminability between pirated and independent models, there is no assessment of scenarios with "unknown" models or more ambiguous conditions (e.g., surrogate attacks specifically designed to evade fingerprints; adaptive adversaries targeting decision geometry). Similarly, detailed analysis of the potential for false positives in more open-world settings is missing.
>
> **Response to concern 1:**
> We would like to first request the reviewer to clarify the precise definitions of "unknown models'' and "open-world settings'', to ensure that our response is perfectly aligned with his/her concern.
>
> First, based on our understanding, our experimental setup already reflects an open-world scenario: the 120 pirated models and 120 independent models used for performance testing are all unseen/unknown during the fingerprint generation process, and they have no overlap with the models in the surrogate pools. Our results demonstrate the fingerprinting method's efficacy for ownership verification across such unknown model instances. If by "unknown'' models you refer to arbitrary models that may even perform different tasks, fingerprinting methods—including ours and all existing adversarial-example-based approaches—naturally reject them. Fingerprints are tied to the output space of the protected model. Thus, if the protected model is a classifier but the suspicious model is, for example, a segmentation network, then feeding the fingerprint samples into the suspicious model produces outputs that are incompatible with the expected classification output format. In this case, the verification score cannot be correctly calculated, indicating that the suspicious model is not a pirated version of the protected model.
>
> Second, regarding "surrogate attacks specifically designed to evade fingerprints and adaptive adversaries targeting decision geometry," we tested comprehensively against a set of six different types of attacks which are common strategies proposed in the literature to circumvent fingerprint verification by shifting decision geometry. If the reviewer has a specific, unaddressed surrogate attack or evasion strategy in mind, we would appreciate further details so we can thoroughly evaluate its impact.
>
> ---
>
> ## Concern 2:
>
> AnaFP's practicality depends on the construction of surrogate pirated and independent pools. While sensitivity to pool size/diversity is empirically analyzed, the method still assumes access to representative pools, which may not always be feasible—especially in dynamic, heterogeneous model deployment ecosystems. There is little discussion of how errors in pool selection could impact the upper/lower bound estimation and thus fingerprint viability.
>
> **Response to concern 2:**
> Our empirical results indicate that AnaFP is highly insensitive to the representativeness of surrogate pools. First, Sec. 5.2.1 shows that even when the surrogate pirated pool contains only two attack types (fine-tuning and KD) and the surrogate independent pool contains only two architectures, AnaFP still generalizes successfully to testing pools with six attacks and 7-10 unseen architectures. This performance sufficiently shows that AnaFP achieves effective fingerprinting performance even when the surrogate models are not fully representative.
>
> Additionally, we would appreciate it if the reviewer could clarify what is specifically meant by a "dynamic, heterogeneous model deployment ecosystem'' and in what way such a scenario might affect DNN fingerprinting. This would allow us to better understand and address the concern with appropriate technical depth.
>
> ---
>
> ## Concern 3:
>
> Although quantile-based relaxation is well motivated, the chosen quantiles are somewhat arbitrary and task dependent. The method remains sensitive to settings that are not fully justified theoretically.
>
> **Response to concern 3:**
> We would like to highlight that Table 3 in Section 5.2.2 clearly shows the insensitivity of AnaFP’s verification performance to the precise choice of quantile thresholds. Once the quantiles are relaxed to a level that produces a number of fingerprints (e.g., $>$50), the resulting AUC remains stable over a wide range of settings. In other words, quantile-based relaxation is a practical estimation step rather than a fragile hyperparameter. We hope this clarification can fully address the reviewer's concern.

---

> > ### Author Response · Authors · 2025-11-21
> >
> > ---
> >
> > ## Concern 4:
> >
> > The fingerprint creation protocol may incur significant computational costs for large-scale or production systems. There is no empirical profiling of runtime or scalability as the number of fingerprints, surrogates, or model size increases.
> >
> > **Response to concern 4:**
> > We profiled the runtime of the three steps in our proposed fingerprinting method: Step 1 (anchor selection), Step 2 (adversarial perturbation computation), and Step 3 (searching stretch factors). The results are presented in the following table. From the table, we can see that these runtime costs are acceptable. Specifically, first, Step 2 dominates the total runtime, requiring 1000–3800 seconds depending on the protected model size and surrogate pool size, whereas the time cost of Step 1 is negligible, and Step 3 contributes a moderate amount (1000–2100 seconds). Second, the time cost scales with model size: fingerprinting ViT-S/16 requires significantly more time cost the an fingerprinting ResNet-18, because a larger model incurs more computations during the fingerprint generation process. Third, increasing the surrogate pool from 4 to 8 models increases the runtime of Step 3, where grid search accounts for the major time cost. Overall, the runtime is moderate. Particularly, we can further reduce the runtime by using faster adversarial attacks (e.g., PGD/DeepFool) and more efficient $\tau$-search strategies. We will incorporate the discussion in the revised version.
> >
> > **Time cost for the three fingerprint-generation steps: Step 1 (anchor selection), Step 2 (adversarial perturbation generation), and Step 3 ($\tau$ search).**
> >
> > | Protected Model | Surrogate pool size | Step 1 | Step 2 | Step 3 |
> > | --------------- | ------------------- | ------ | ------ | ------ |
> > | ViT-S/16        | 4                   | 63s    | 3856s  | 1276s  |
> > | ResNet18        | 4                   | 8s     | 1004s  | 976s   |
> > | ViT-S/16        | 8                   | 61s    | 3855s  | 2121s  |
> > | ResNet18        | 8                   | 8s     | 997s   | 1419s  |
> >
> > ---
> >
> > ## Question 1:
> >
> > How would AnaFP fare against adaptive attackers who are aware of the fingerprint generation protocol and attempt to evade by specifically distorting or randomizing localized decision boundaries? Are there (perhaps adversarially optimized) model modifications that could reduce the efficacy of the derived interval without violating standard performance constraints?
> >
> > **Response to question 1:**
> > Regarding the "fingerprint-aware'' adaptive attackers, there are two different threats. We describe them as follows.
> >
> > * One is that the attacker only knows that our fingerprinting method is the adversarial-example-based, and applies adversarial training to reshape local decision boundaries. This threat has already been taken into account in our paper, and experimental results have shown that AnaFP outperforms existing methods under the adversarial training attack. This is because our fingerprints lie at locations with distances from the boundaries which maximize the robustness against boundary shifts.
> > * The other threat is that the attacker has direct access to the fingerprint samples themselves and fine-tunes the stolen model to override its behavior on these exact inputs. To the best of our knowledge, no adversarial-example-based fingerprinting scheme can defend against this attack, as it is easy to effectively overwrites the model’s behavior on a finite set of known samples. In practice, fingerprint samples are typically kept confidential by the model owner or shared with trusted third-party.
> >
> > ---
> >
> > ## Question 2:
> >
> > Can the authors clarify or provide more guidance on choosing quantile thresholds for the relaxation strategy (see Table 3), beyond the trial-and-error or “reasonable number of fingerprints” heuristic? Any analytical or automated procedure to further reduce parameter sensitivity?
> >
> > **Response to question 2:**
> > Table 3 in Section 5.2.2 reveals a clear empirical trend: as quantile thresholds are relaxed, both the number of valid fingerprints and the corresponding AUC increase, until performance stabilizes once approximately 30–50 valid fingerprints are obtained. Beyond this point, further relaxation yields marginal benefits, indicating a performance plateau. This observation motivates a straightforward yet effective automation strategy: incrementally relax the quantile threshold until a sufficient number of valid fingerprints (e.g., 50) are identified.

---

> ### Author Response · Authors · 2025-11-21
>
> ---
>
> ## Question 3:
>
> The use of grid search with potentially many anchors and surrogate models may be computationally expensive, especially for large-scale or production systems. Could you provide more details on the runtime complexity and memory requirements for AnaFP in realistic settings?
>
> **Response to question 3:**
>
> We profiled the runtime of the three steps in our proposed fingerprinting method: Step 1 (anchor selection), Step 2 (adversarial perturbation computation), and Step 3 (searching stretch factors). The results are presented in the following table. From the table, we can see that these runtime costs are acceptable. Specifically, first, Step 2 dominates the total runtime, requiring 1000–3800 seconds depending on the protected model size and surrogate pool size, whereas the time cost of Step 1 is negligible, and Step 3 contributes a moderate amount (1000–2100 seconds). Second, the time cost scales with model size: fingerprinting ViT-S/16 requires significantly more time cost than fingerprinting ResNet-18, because a larger model incurs more computations during the fingerprint generation process. Third, increasing the surrogate pool from 4 to 8 models increases the runtime of Step 3, where grid search accounts for the major time cost. Overall, the runtime is moderate. Particularly, we can further reduce the runtime by using faster adversarial attacks (e.g., PGD/DeepFool) and more efficient $\tau$-search strategies
>
> **Time cost for the three fingerprint-generation steps: Step 1 (anchor selection), Step 2 (adversarial perturbation generation), and Step 3 ($\tau$ search).**
>
> | Protected Model | Surrogate pool size | Step 1 | Step 2 | Step 3 |
> | --------------- | ------------------- | ------ | ------ | ------ |
> | ViT-S/16        | 4                   | 63s    | 3856s  | 1276s  |
> | ResNet18        | 4                   | 8s     | 1004s  | 976s   |
> | ViT-S/16        | 8                   | 61s    | 3855s  | 2121s  |
> | ResNet18        | 8                   | 8s     | 997s   | 1419s  |
>
> We also measured GPU memory usage during Step 2 and Step 3, the experimental results are summarized in the following table. We can observe that first, Step 2 requires the highest memory because it performs adversarial optimization, consuming 13 GB for ResNet-18 and about 40 GB for ViT-S/16, whereas Step 3 requires 1–3 GB. Second, memory usage grows with model size: ViT-S/16 on Tiny-ImageNet has roughly three times the memory requirements of ResNet-18 due to a larger model size. Third, increasing the surrogate pool size from 4 to 8 models increases memory usage slightly.
>
> **Peak memory consumption for Step 2 (adversarial perturbation generation) and Step 3 (grid search).**
>
> | Protected Model | Surrogates | Step 2 | Step 3 |
> | --------------- | ---------- | ------ | ------ |
> | ViT-S/16        | 4          | 40G    | 3G     |
> | ResNet18        | 4          | 13G    | 1G     |
> | ViT-S/16        | 8          | 41G    | 3.2G   |
> | ResNet18        | 8          | 13G    | 1.1G   |
>
> We will include the profiling of time cost/memory usage and discuss potential efficiency improvement in the revised version.
>
> ---
>
> ## Question 4:
>
> The evaluation focuses on known model modification attacks (pruning, fine-tuning, KD, AT). Have you considered or tested robustness against adaptive attackers who might specifically attempt to evade fingerprinting by perturbing near-anchor regions or strategically altering decision boundaries?
>
> **Response to question 4:**
> Regarding the "fingerprint-aware'' adaptive attackers, there are two different threats. We have discussed in Question 1.
>
> Regarding perturbing “near-anchor’’ regions, it is ineffective because the actual fingerprints are not located near the anchors; they lie near the protected model’s boundary after stretching.
>
> ---
>
> ## Question 5:
>
> While AnaFP shows strong results on CNNs, MLPs, and GNNs, are there any specific architectural choices, dataset characteristics, or domains where the approach might face limitations? For example, very large-scale models or models trained with self-supervised or unsupervised objectives?
>
> **Response to question 5:**
> We have not observed architecture- or dataset-specific failure modes across CNNs, MLPs, and GNNs. However, a potential limitation is the computational cost, particularly for very large-scale models. To address this concern, we can reduce the computational cost by exploiting some speedup techniques, e.g., replacing C&W with more efficient adversarial methods, such as PGD and DeepFool, and designing a coarse-to-fine $\tau$-search method to replace the grid search method. We will discuss this in the revised version.
>
> Regarding self-supervised or unsupervised models, our method relies on decision boundaries and predictive logits, making it naturally suitable for classification tasks.

---

> > ### Comment · Reviewer_ZKy4 · 2025-11-26
> >
> > Thanks for your response. While some concerns have been addressed, several issues remain insufficiently justified. I would raise my score if these issues can be further clarified.
> >
> > ## Regarding Concern 1
> > 1. On “unknown models” and “open-world settings”.
> > Your response interprets “unknown models” purely as unseen instances performing the same task, but the concern is broader: (1) Models trained on similar data but with different objectives; (2) Models architecturally similar but with different output semantics; (3) Models produced by transfer learning from the protected model but diverged significantly. Simply stating that task-mismatched models “naturally reject” fingerprints does not fully address ambiguity in same-task but fundamentally different models.
> > Overall, more explicit discussion is required to clearly delineate threat model boundaries and conditions where AnaFP may fail.
> >
> > ## Regarding Question 1
> > The threat model distinction is reasonable, but two issues remain:
> > 1. Adaptive boundary shaping.
> > Your claim that fingerprints lie at “maximally robust distance from boundaries” requires more justification. (1) Can an adaptive attacker; (2) locally linearize boundaries; (3) apply curvature regularization, or enforce Lipschitz constraints to distort the near-boundary geometry and reduce fingerprint utility?
> >
> > 2. Fingerprint exposure attack.
> > While it is true that no adversarial-example-based fingerprint can defend against full fingerprint exposure, you should explicitly clarify this as a fundamental limitation of the entire methodology, not just your method.
> >
> > ## Regarding Question 2
> > Your proposed heuristic—relaxing quantiles until ~50 fingerprints emerge—is reasonable but remains heuristic. Two follow-up points:
> > 1. Could you design an adaptive stopping criterion based on: (1) stability of interval width, (2) statistical variance across surrogate models, (3) or monotonicity of AUC gains?
> >
> > 2. What prevents an attacker from inducing distributional drift such that quantile-based selection becomes unstable?

---

> > > ### Author Response · Authors · 2025-12-02
> > >
> > > ---
> > >
> > > ## Concern 1
> > >
> > > On “unknown models” and “open-world settings”. Your response interprets “unknown models” purely as unseen instances performing the same task, but the concern is broader:
> > >
> > > 1. Models trained on similar data but with different objectives;
> > > 2. Models architecturally similar but with different output semantics;
> > > 3. Models produced by transfer learning from the protected model but diverged significantly.
> > >
> > > Simply stating that task-mismatched models “naturally reject” fingerprints does not fully address ambiguity in same-task but fundamentally different models. Overall, more explicit discussion is required to clearly delineate threat model boundaries and conditions where AnaFP may fail.
> > >
> > > **Response 1:**
> > >
> > > Thanks for outlining these three specific scenarios regarding “unknown models.” This breakdown helps us delineate the boundary conditions of AnaFP more precisely.
> > >
> > > **(1) Regarding “Models trained on similar data but with different objectives”:**
> > >
> > > *AUCs achieved by AnaFP and baselines under two datasets, where independent models are trained with a similar dataset to the protected model but with a different training objective.*
> > >
> > > | Method      | CNN/CIFAR-10  | CNN/CIFAR-100 |
> > > | ----------- | ------------- | ------------- |
> > > | AnaFP       | 0.934 ± 0.003 | 0.875 ± 0.008 |
> > > | UAP         | 0.838 ± 0.011 | 0.793 ± 0.025 |
> > > | IPGuard     | 0.689 ± 0.053 | 0.697 ± 0.071 |
> > > | MargnFinger | 0.690 ± 0.077 | 0.624 ± 0.066 |
> > > | AKH         | 0.685 ± 0.036 | 0.773 ± 0.042 |
> > > | ADV-TRA     | 0.877 ± 0.014 | 0.840 ± 0.028 |
> > > | GMFIP       | 0.806 ± 0.040 | 0.766 ± 0.064 |
> > >
> > > We additionally evaluated whether AnaFP can reliably distinguish the pirated models from independent models that are trained on a similar dataset as the protected model but with a different objective. Concretely, the protected models are trained using the standard cross-entropy loss, whereas all independent models are trained using the label-smoothing cross-entropy loss function. The results in the table above show that AnaFP continues to achieve strong discrimination performance, obtaining the highest AUC on both CIFAR-10 (0.934 ± 0.003) and CIFAR-100 (0.875 ± 0.008). These results indicate that AnaFP maintains discriminability even when independent models are trained on similar data but with a different training objective from the protected model.
> > >
> > > **(2) Regarding “Models architecturally similar but with different output semantics”:**
> > > If the scenario raised by the reviewer refers to: an independent model shares the exact architecture, input, and output dimension with the protected model, but possesses completely different output semantics (e.g., the protected model classifies (N) animals, an independent model classifies (N) digits). AnaFP is robust to this semantic mismatch because our verification protocol relies on the model’s ultimate classification label, rather than the raw logit distribution. When the fingerprints are fed into the independent model, whose underlying semantics are entirely different from the protected model, the independent model’s output label will fail to match the expected fingerprint label, ensuring its correct rejection and effectively maintaining a low false positive rate.
> > >
> > > **(3) “Models produced by transfer learning but diverged significantly”:**
> > > We accept this scenario as a fundamental limitation of our method and most existing fingerprinting methods, which we will explicitly clarify in the Discussion section (Appendix) of the revised manuscript. For the transfer learning case, if a pirated model is heavily fine-tuned for a completely different task such that the model diverges significantly, or if the output format itself changes (e.g., shifting from classification to segmentation), the ownership traces (fingerprints) tied to the original model’s specific functional geometry will inevitably be lost. Therefore, this reflects a limitation of the existing fingerprinting method, including AnaFP, when the protected model has been dramatically altered.
> > >
> > > ---

---

> > > ### Author Response · Authors · 2025-12-02
> > >
> > > ## Question 1
> > >
> > > The threat model distinction is reasonable, but two issues remain:
> > >
> > > Adaptive boundary shaping. Your claim that fingerprints lie at “maximally robust distance from boundaries” requires more justification. Can an adaptive attacker locally linearize boundaries? Apply curvature regularization or Enforce Lipschitz constraints to distort the near-boundary geometry and reduce fingerprint utility?
> > >
> > > Fingerprint exposure attack. While it is true that no adversarial-example-based fingerprint can defend against full fingerprint exposure, you should explicitly clarify this as a fundamental limitation of the entire methodology, not just your method.
> > >
> > > **Response to Question 1:**
> > >
> > > **For adaptive boundary shaping.** When the fingerprints are not exposed, the attacker does not know which specific boundary segments contribute to the fingerprint signal. Therefore, to reliably suppress potential fingerprint regions, the attacker cannot restrict manipulation to a single localized patch; instead, they must modify the decision boundary across many regions of the input space. Local linearization techniques can, in principle, reshape the boundary in a small neighborhood, but without knowing where the fingerprints lie, the attacker would need to apply such operations repeatedly over a wide range of inputs. Curvature regularization and Lipschitz constraints are even broader in nature, as they directly encourage globally or semi-globally smoother decision surfaces. Applying these mechanisms with sufficient strength to cover all possible fingerprint locations inevitably alters the model’s behavior on many natural samples, leading to noticeable accuracy degradation.
> > >
> > > In addition, AnaFP places fingerprints not directly on the decision boundary but at margin-controlled regions. This design makes small, localized boundary shifts insufficient: to neutralize such fingerprints, the attacker would have to displace the boundary deeply enough to cross into these interior margin zones. Achieving such a displacement requires more substantial geometric modifications, affecting a larger neighborhood of natural samples and further increasing the accuracy cost for the attacker.
> > >
> > > **For the fingerprint exposure attack.** Thanks for pointing it out, we will explicitly clarify in the revised version that fingerprint-exposure attacks represent a fundamental, paradigm-level limitation of model fingerprinting, and are not specific to the proposed method.

---

> > > ### Author Response · Authors · 2025-12-02
> > >
> > > ---
> > >
> > > ## Question 2
> > >
> > > Your proposed heuristic—relaxing quantiles until ~50 fingerprints emerge—is reasonable but remains heuristic. Two follow-up points:
> > >
> > > 1. Could you design an adaptive stopping criterion based on stability of interval width, statistical variance across surrogate models, or monotonicity of AUC gains?
> > > 2. What prevents an attacker from inducing distributional drift such that quantile-based selection becomes unstable?
> > >
> > > **Response to Question 2**
> > >
> > > **Point 1:**
> > > We thank the reviewer for suggesting several possible adaptive stopping criteria.
> > >
> > > **(1) For the stability of the interval width.**
> > > In AnaFP, we use the relaxed parameter estimates to obtain the admissible interval, and relaxing quantiles can enlarge the interval monotonically, which does not create any “stability” that can be used as a stopping rule. Thus, an interval-width stability criterion is not compatible with AnaFP.
> > >
> > > **(2) For the statistical variance across surrogate models.**
> > > In AnaFP, surrogate models are not used to provide per-model estimates. Instead, all surrogates are collectively used for estimating three global worst-case quantities used in the analytical bounds, i.e., (m_{\min}), (L_{\text{uniq}}), and (\epsilon_{\text{logit}}). These quantities are derived directly from order statistics (e.g., minimum, maximum, or relaxed quantiles) over all the surrogate pirated models or all the surrogate independent models. Because we do not compute per-surrogate estimates and do not aggregate them across models, there is no “statistical variance across surrogate models” in our pipeline. Therefore, a variance-based stopping rule is not applicable to how AnaFP uses surrogate pools.
> > >
> > > **(3) For the monotonicity of AUC gains.**
> > > This idea is feasible in principle. If an additional evaluation model set—with both pirated and independent models—is available, one could gradually relax the quantile and monitor the resulting AUC on this set, stopping when further relaxation no longer improves performance. Such a procedure would provide a more direct and potentially more effective quantile selection criterion. However, it also requires constructing an evaluation set large enough to yield reliable AUC estimates, which introduces substantial additional training cost. Our current approach instead uses the number of valid fingerprints as a lightweight proxy to avoid this overhead, even though it is less direct than an AUC-based evaluation. It is noteworthy that the surrogate models used in AnaFP cannot serve as this evaluation set, because they are directly used for estimating the parameters that determine the admissible intervals; reusing them for AUC evaluation would yield an overly optimistic assessment.
> > >
> > > **Point 2:**
> > > In AnaFP, quantile-based relaxation is performed once during fingerprint generation on the owner side, using only the protected model and the surrogate pools. These quantile thresholds determine the admissible interval and remain fixed afterward. An attacker modifying a stolen model cannot retroactively change these thresholds or influence how the quantiles were computed. The attacker can only alter how the stolen model behaves. If an extremely strong attack produces a pirated model significantly diverged from the protected model, this affects verification performance itself, but does not destabilize the quantile-selection procedure. In other words, the quantile selection is fully determined before any attacker action, and distributional drift from the attacker does not enter into the quantile-estimation process.

---

### Official Review · Reviewer_JWiN · 2025-11-01

**Soundness:** 4
**Presentation:** 3
**Contribution:** 3
**Rating:** 6
**Confidence:** 4

**Summary:**

This paper presents AnaFP, an analytical framework for generating adversarial fingerprint samples to verify the ownership of deep neural networks. Unlike prior heuristic approaches that place fingerprints near decision boundaries, AnaFP derives theoretical upper and lower bounds on the fingerprint-to-boundary distance, governed by a scaling factor. The bounds are motivated by two conflicting objectives: robustness against model modifications and uniqueness across independently trained models. The authors further propose a practical approximation using surrogate model pools and quantile relaxation, followed by a grid search to select feasible fingerprints.

**Strengths:**

1. The paper provides a clean analytical formulation for the long-standing heuristic choice of how far fingerprints should lie from the decision boundary. By deriving explicit upper and lower bounds on the scaling factor, the authors turn this into a principled optimization problem rather than an empirical guess.
2. Experiments cover multiple model families (CNN/MLP/GNN), datasets (image and graph), a wide range of model modification attacks, several baselines, and repeated runs to report mean ± std. Tables and figures indicate consistently strong AUCs for AnaFP.

**Weaknesses:**

1. The related-work section lacks a systematic overview of existing fingerprinting or watermarking techniques for deep models.
2. While the appendix mentions some hyperparameters, key details such as exact model architectures used and number of models in each surrogate pool are missing or only briefly listed. These are central to understanding the experiment setup. Important configurations should appear in the main text, not just in appendices.
3. While AnaFP is analytically grounded, its deployment requires maintaining surrogate model pools, performing adversarial optimization (e.g., C&W attacks) for fingerprint generation, and running grid search for each anchor. These steps incur substantial training and inference cost, which may limit practicality in large-scale or resource-constrained settings. The paper would benefit from reporting quantitative cost and discussing potential efficiency improvements.

**Questions:**

See Weaknesses

---

> ### Author Response · Authors · 2025-11-21
>
> ## Concern 1:
>
> The related-work section lacks a systematic overview of existing fingerprinting or watermarking techniques for deep models.
>
> **Response to concern 1:**
> Our paper primarily focuses on the uniqueness and robustness of adversarial-example-based fingerprinting that is applied for classification tasks. Hence, we provide a comprehensive survey on various adversarial-example-based fingerprinting methods. We agree with the reviewer that discussing more fingerprinting methods that are not adversarial-example-based can help readers understand general fingerprinting techniques. Thus, we will provide a systematic overview of existing fingerprinting techniques in the revised version.
>
> ---
>
> ## Concern 2:
>
> While the appendix mentions some hyperparameters, key details such as exact model architectures used and number of models in each surrogate pool are missing or only briefly listed. These are central to understanding the experiment setup. Important configurations should appear in the main text, not just in appendices.
>
> **Response to concern 2:**
> We thank the reviewer for the suggestion. In the revision, we will move the key settings—including the model architectures used in the surrogate pools and the number of models in each pool—from the appendix into the main text to make the experimental setup clearer and more complete.
>
> ---

---

> ### Author Response · Authors · 2025-11-21
>
> ## Concern 3:
>
> While AnaFP is analytically grounded, its deployment requires maintaining surrogate model pools, performing adversarial optimization (e.g., C&W attacks) for fingerprint generation, and running grid search for each anchor. These steps incur substantial training and inference cost, which may limit practicality in large-scale or resource-constrained settings. The paper would benefit from reporting quantitative cost and discussing potential efficiency improvements.
>
> **Response to concern 3:**
> We profiled the runtime of the three steps in our proposed fingerprinting method: Step 1 (anchor selection), Step 2 (adversarial perturbation computation), and Step 3 (searching stretch factors). The results are presented in the following table. From the table, we can see that these runtime costs are acceptable. Specifically, first, Step 2 dominates the total runtime, requiring 1000–3800 seconds depending on the protected model size and surrogate pool size, whereas the time cost of Step 1 is negligible, and Step 3 contributes a moderate amount (1000–2100 seconds). Second, the time cost scales with model size: fingerprinting ViT-S/16 requires significantly more time cost than fingerprinting ResNet-18, because a larger model incurs more computations during the fingerprint generation process. Third, increasing the surrogate pool from 4 to 8 models increases the runtime of Step 3, where grid search accounts for the major time cost. Overall, the runtime is moderate. Particularly, we can further reduce the runtime by using faster adversarial attacks (e.g., PGD/DeepFool) and more efficient $\tau$-search strategies.
>
> **Time cost for Step 1, Step 2, and Step 3**
>
> | Protected Model | Surrogate pool size | Step 1 | Step 2 | Step 3 |
> | --------------- | ------------------- | ------ | ------ | ------ |
> | ViT-S/16        | 4                   | 63s    | 3856s  | 1276s  |
> | ResNet18        | 4                   | 8s     | 1004s  | 976s   |
> | ViT-S/16        | 8                   | 61s    | 3855s  | 2121s  |
> | ResNet18        | 8                   | 8s     | 997s   | 1419s  |
>
> We also measured GPU memory usage during Step 2 and Step 3, the experimental results are summarized in the following table. We can observe that first, Step 2 requires the highest memory because it performs adversarial optimization, consuming 13 GB for ResNet-18 and about 40 GB for ViT-S/16, whereas Step 3 requires 1–3 GB. Second, memory usage grows with model size: ViT-S/16 on Tiny-ImageNet has roughly three times the memory requirements of ResNet-18 due to a larger model size. Third, increasing the surrogate pool size from 4 to 8 models increases memory usage slightly.
>
> **Peak memory consumption for Step 2 and Step 3**
>
> | Protected Model | Surrogates | Step 2 | Step 3 |
> | --------------- | ---------- | ------ | ------ |
> | ViT-S/16        | 4          | 40G    | 3G     |
> | ResNet18        | 4          | 13G    | 1G     |
> | ViT-S/16        | 8          | 41G    | 3.2G   |
> | ResNet18        | 8          | 13G    | 1.1G   |
>
> The efficiency of AnaFP can be further improved through the following practices. First, regarding surrogate pools, independent surrogate models can be drawn directly from publicly available models, eliminating the need for additional training, while generating pirated surrogate models via fine-tuning or pruning incurs very low cost. Even knowledge distillation, which is relatively more expensive, remains cheaper than training models from scratch. Second, the adversarial example generation step can be replaced with more efficient alternatives of C&W, such as PGD or DeepFool, which can reduce computations. Third, we think that a coarse-to-fine search strategy could provide better efficiency than naive grid search. We will incorporate the profiling of time cost/memory usage and a discussion on potential efficiency improvements in the revised version.

---

### Official Review · Reviewer_5yER · 2025-11-03

**Soundness:** 3
**Presentation:** 3
**Contribution:** 3
**Rating:** 6
**Confidence:** 4

**Summary:**

This paper focuses on the fingerprint generation for model ownership verification based on two essential properties—robustness and uniqueness from a theoretical perspective, which links the two constraints introduced by robustness and uniqueness to the fingerprint-to-boundary distance. The authors formalize two key properties—robustness (against model modifications) and uniqueness (against independently trained models)—and derive τ bounds that satisfy both. They further relax these bounds via surrogate model pools and quantile estimation, and determine τ via grid search. Extensive experiments show superior AUC over prior work across CNN, MLP, and GNN architectures.

**Strengths:**

1.	To the best of my knowledge, this might be the first work to analytically characterize the admissible τ-interval that jointly guarantees robustness and uniqueness. The derivation is clean and verifiable, and surrogate pools + quantile relaxation elegantly make the bounds estimable without violating the theory.
2.	Consistently highest AUC on all six modification attacks and four datasets, while showing low sensitivity to pool size/quantile settings.
3.	Extensive results demonstrate the superior performance of AnaFP over other methods, which can generalize to CNN, MLP, and GNN models among diverse model IP attacks.
4.	While casting fingerprint generation as a “distance-to-boundary” control problem is not new (MarginFinger, IPGuard), AnaFP’s main novelty is adding analytical bounds on the stretch factor τ. I believe this idea and perspective could motivate more follow-up work in the community.

**Weaknesses:**

1.	My first concern is about the cost. Every anchor requires a full targeted C&W optimization (≈ 3,000 steps) and a 500-point grid search over τ. Complexity is O(N_f × 3,000 × 500) forward-backward passes—impractical for ImageNet-scale models. No speed-up (e.g., early termination, bisection search, Jacobian-free solvers) is discussed.
2.	Missing baselines. The experimental comparison is restricted to adversarial-example methods; recent non-adversarial ownership schemes are omitted, which can provide more necessary information on AnaFP’s superior performance.
3.	The evaluated surrogate piracy pool is limited to six handcrafted attacks (fine-tune, KD, prune). Stronger attack combinations (like prune→KD or adversarial training with fingerprint-aware data) are not covered.

**Questions:**

1.	Could the authors provide more information on the fingerprints’ generation cost rather than only the detection performance? Like time and memory size compared to other existing methods
2.	KD is known as the most challenging IP attack in the real-world scenario. I wonder about the detailed attack settings and detection performance(e.g., detection rate and FPR) on this attack, which could significantly help readers to buy these results.

Overall, although the current version of this paper lacks some important results and details, I believe the theoretical perspective presented is somewhat reasonable and could inspire more future work in similar directions. Therefore, I am currently slightly inclined to accept it. I hope the author will provide point-by-point responses to the issues I raised during the rebuttal phase. If these issues can be adequately clarified, I would be willing to raise my score and support the acceptance of this paper.

---

> ### Author Response · Authors · 2025-11-21
>
> ---
>
> ## Concern 1:
>
> My first concern is about the cost. Every anchor requires a full targeted C&W optimization (around 3,000 steps) and a 500-point grid search over $\tau$. Complexity is O($N_f$ × 3,000 × 500) forward-backward passes—impractical for ImageNet-scale models. No speed-up (e.g., early termination, bisection search, Jacobian-free solvers) is discussed.
>
> **Response to concern 1:**
> Thanks for your constructive suggestions.
>
> First, we would like to clarify that for each anchor, the targeted C&W optimization and the subsequent $\tau$ search are {sequential} rather than multiplicative, thus resulting in the computational cost of approximately $O(N_f \times (3000 + 500))$, instead of $O(N_f \times 3000 \times 500)$. In particular, the C&W optimizer does not always reach the maximum of 3,000 iterations: it will be terminated once a valid perturbation is found, leading to lower cost in practical systems.
>
> Second, regarding speed-ups, we will add the discussion in the revised version.
>
> * The C&W optimization in the Computing Minimal Decision-altering Perturbation Step (Step 2) can be replaced by faster adversarial solvers, such as PGD, DeepFool, and even FGSM-style methods. These solvers can further reduce the computations. We will include the discussion in the revised version. However, it is worth nothing that using them or the C&W optimization does not affect the core idea of our method.
> * For searching $\tau$, our current implementation uses a uniform grid search for simplicity. A more efficient coarse-to-fine search is certainly possible to reduce the computation. Methods like the bisection search, however, cannot be directly applied. This is because the relationship between $\tau$ and $\tau_{\text{lower}}(\tau)$ is not unimodal or monotonic, which are typically required for binary search-based methods to work effectively.
>
> ---
>
> ## Concern 2:
>
> Missing baselines. The experimental comparison is restricted to adversarial-example methods; recent non-adversarial ownership schemes are omitted, which can provide more necessary information on AnaFP’s superior performance.
>
> **Response to Concern 2:**
> First, we would like to clarify that in our current version, we have already considered comparing with AKH, a recently-developed non-adversarial-example-based fingerprinting method that constructs fingerprints from the protected model’s naturally misclassified samples.
>
> Second, to further validate AnaFP's performance, in the revised version, we will further consider GMFIP and ADV-TRA. GMFIP, a newly-proposed non-adversarial-example-based fingerprinting method, first trains a generator to synthesize fingerprint samples capturing the source model’s unique decision characteristics, and then includes a refinement stage and a binary classifier to select high-quality fingerprints while preserving the model owner’s utility. ADV-TRA, an adversarial-example-based fingerprinting method, constructs adversarial trajectories that traverse different regions of the decision boundary, instead of individual adversarial examples, to achieve strong verification performance.
>
> **AUCs achieved by different approaches across DNN models and datasets**
>
> | Method       | CNN (CIFAR-10)    | CNN (CIFAR-100)   | MLP (MNIST)       | GNN (PROTEINS)    |
> | ------------ | ----------------- | ----------------- | ----------------- | ----------------- |
> | AnaFP (ours) | **0.957 ± 0.002** | **0.893 ± 0.005** | **0.963 ± 0.002** | **0.926 ± 0.005** |
> | UAP          | 0.850 ± 0.010     | 0.806 ± 0.021     | 0.906 ± 0.004     | --                |
> | IPGuard      | 0.715 ± 0.075     | 0.725 ± 0.090     | 0.873 ± 0.018     | 0.636 ± 0.067     |
> | MarginFinger | 0.671 ± 0.064     | 0.630 ± 0.072     | 0.653 ± 0.051     | --                |
> | AKH          | 0.723 ± 0.016     | 0.802 ± 0.019     | 0.851 ± 0.013     | 0.854 ± 0.021     |
> | ADV-TRA      | 0.878 ± 0.012     | 0.850 ± 0.024     | 0.887 ± 0.005     | --                |
> | GMFIP        | 0.814 ± 0.047     | 0.781 ± 0.075     | 0.892 ± 0.023     | --                |
>
> The updated experimental results are summarized in the table above. From this table, we can see that our proposed AnaFP consistently achieves the best AUC across all architectures and datasets. Although AKH, GMFIP, and ADV-TRA show relatively strong performance across most settings, their performance is worse than AnaFP, which corroborates the effectiveness of our proposed AnaFP.

---

> > ### Author Response · Authors · 2025-11-21
> >
> > ---
> >
> > ## Concern 3:
> >
> > The evaluated surrogate piracy pool is limited to six handcrafted attacks (fine-tune, KD, prune). Stronger attack combinations (like prune→KD or adversarial training with fingerprint-aware data) are not covered.
> >
> > **Response to concern 3:**
> > First of all, we would like to clarify that the surrogate piracy pool is used for estimating theoretical bounds, whereas both the pirated and independent model sets are used for experimentally evaluating AnaFP's performance.
> >
> > Second, two attack types (fine-tuning and knowledge distillation) for each task, as stated in Appendix C.2.2, are considered when constructing the surrogate piracy pool, while six attack types are considered to construct the pirated model set for each task.
> >
> > Third, we evaluate the impact of the surrogate pirated and independent model pools (including the pool size and pool diversity) on AnaFP's performance in Section 5.2.1. To assess the pool diversity, we consider low-diversity setting with one attack type (fine-tuning), medium-diversity setting (fine-tuning and knowledge distillation), and high-diversity setting (fine-tuning, knowledge distillation, and the combination of noise injection and fine-tuning). The experimental results demonstrate that even when the surrogate piracy pool is constructed by considering the two attack types, the learned fingerprints achieve superior performance when they are tested using the pirated and independent model sets, indicating its strong generalization.
> >
> > Fourth, we further consider a stronger attack combination--prune-KD attack--which is suggested by the reviewer--for constructing the pirated model set, and assess the performance of AnaFP by comparing with baselines. The experimental results in the CNN/CIFAR-10 setting are summarized below:
> >
> > **AUCs of different fingerprinting methods under the prune–KD attack on CNN/CIFAR-10**
> >
> > | Method       | AUC (CNN/CIFAR-10) |
> > | ------------ | ------------------ |
> > | AnaFP (ours) | **0.689 ± 0.031**  |
> > | UAP          | 0.625 ± 0.026      |
> > | IPGuard      | 0.596 ± 0.079      |
> > | MarginFinger | 0.543 ± 0.088      |
> > | AKH          | 0.636 ± 0.047      |
> > | ADV-TRA      | 0.633 ± 0.035      |
> > | GMFIP        | 0.601 ± 0.068      |
> >
> > Fifth, regarding the fingerprint-aware attack, there are two different threats. One is that the attacker only knows that our fingerprinting method is the adversarial-example-based, and applies adversarial training to reshape local decision boundaries. This threat has already been taken into account in our paper, and experimental results have shown that AnaFP outperforms existing methods under the adversarial training attack. The other threat is that the attacker has direct access to the fingerprint samples themselves and fine-tunes the stolen model to override its behavior on these exact inputs. To the best of our knowledge, no adversarial-example-based fingerprinting scheme can defend against this attack, as it is easy to effectively overwrites the model’s behavior on a finite set of known samples. In practice, fingerprint samples are typically kept confidential by the model owner or shared with trusted third-party.

---

> > > ### Author Response · Authors · 2025-11-21
> > >
> > > ---
> > >
> > > ## Question 1:
> > >
> > > Could the authors provide more information on the fingerprints’ generation cost rather than only the detection performance? Like time and memory size compared to other existing methods
> > >
> > > **Response to question 1:**
> > > Thanks for your constructive suggestion. We conduct the experiment and compare the computing time and peak GPU memory consumed by AnaFP and baselines, which will be incorporated into the revised version. The experimental results are summarized in the table below. Specifically, AKH and IPGuard are the most lightweight methods: AKH operates purely through forward inference, and IPGuard performs small-batch gradient updates without minimizing perturbation magnitude, resulting in runtimes of around one minute and memory usage below 3 GB. UAP is the most expensive method, as it requires multiple full passes over the training data to optimize a universal perturbation, leading to high computing time and memory usage. Both MarginFinger and GMFIP achieve moderate cost by training a generator model to synthesize boundary-adjacent samples. ADV-TRA generates adversarial trajectories as fingerprints, which causes a large time cost due to the sequential generation process. Compared to these baselines, AnaFP consumes the highest memory but achieves a balanced runtime. More specifically, AnaFP costs the most GPU memory—up to 40 GB on ViT-S/16, as it processes a large batch of anchors in parallel during the adversarial optimization step. This parallel design reduces the total number of optimization iterations, thus reducing the overall runtime. Despite the high memory usage, AnaFP can still run on a single A100 GPU, making the cost manageable in practice.
> > >
> > > **Time cost and peak GPU memory consumption of different fingerprinting methods**
> > >
> > > | Method       | ResNet-18 Time | ResNet-18 Mem | ViT-S/16 Time | ViT-S/16 Mem |
> > > | ------------ | -------------- | ------------- | ------------- | ------------ |
> > > | AnaFP (ours) | 33 m 8 s       | 13 GB         | 1 h 26 m      | 40 GB        |
> > > | UAP          | 4 h 55 m       | 4 GB          | 8 h 10 m      | 10 GB        |
> > > | IPGuard      | 1 m 6 s        | 1 GB          | 1 m 41 s      | 3 GB         |
> > > | MarginFinger | 51 m 18 s      | 4 GB          | 1 h 14 m      | 10 GB        |
> > > | AKH          | 20 s           | 1 GB          | 1 m 10 s      | 3 GB         |
> > > | ADV-TRA      | 58 m 01 s      | 4 GB          | 1 h 29 m      | 10 GB        |
> > > | GMFIP        | 1 h 15 m       | 4 GB          | 1 h 40 m      | 10 GB        |
> > >
> > >
> > >
> > > ---
> > >
> > > ## Question 2:
> > >
> > > KD is known as the most challenging IP attack in a real-world scenario. I wonder about the detailed attack settings and detection performance(e.g., detection rate and FPR) on this attack, which could significantly help readers to buy these results.
> > >
> > > **Response to question 2:**
> > > We agree that Knowledge Distillation (KD) is one of the most challenging IP attacks for fingerprinting methods. We employ a classical KD setting in our evaluation: the student model is trained to match the teacher model’s output distribution using the KL divergence loss with temperature $T=1$. This setup is widely used and represents a realistic knowledge-transfer attack.
> > >
> > > In addition to the main metric AUC, which reflects the overall detection capability across all threshold settings, we will also assess TPR, FPR, TNR, and FNR for the KD attack in the revised version, as suggested by the reviewer. The experimental results are summarized in the Table below. From this table, we can observe that AnaFP achieves the strongest separation between pirated models under the KD attack and independent models, reaching a high true positive rate (TPR=0.80) while keeping the false positive rate relatively low (FPR=0.22). ADV-TRA and UAP perform worse than AnaFP but better than other baselines, showing a moderate-level performance (TPR=0.73, FPR=0.30 for UAP, and TPR=0.75, FPR=0.27 for ADV-TRA). In contrast, AKH, IPGuard, GMFIP, and MarginFinger are relatively less capable of distinguishing pirated models under the KD attack and independent models, reflected by their high FPR (0.36-0.47) and only moderately high TPR (0.57-0.67).
> > >
> > >
> > > **Detection performance under the KD attack for CNN/CIFAR10**
> > >
> > > | Method       | TPR      | FPR      | TNR  | FNR  |
> > > | ------------ | -------- | -------- | ---- | ---- |
> > > | AnaFP (ours) | **0.80** | **0.22** | 0.78 | 0.20 |
> > > | UAP          | 0.73     | 0.30     | 0.70 | 0.27 |
> > > | IPGuard      | 0.58     | 0.46     | 0.54 | 0.42 |
> > > | MarginFinger | 0.57     | 0.47     | 0.53 | 0.43 |
> > > | AKH          | 0.63     | 0.40     | 0.60 | 0.37 |
> > > | ADV-TRA      | 0.75     | 0.27     | 0.73 | 0.25 |
> > > | GMFIP        | 0.67     | 0.36     | 0.64 | 0.33 |
> > >
> > > ---

---

### Meta-Review · Area_Chair_zzL5 · 2026-01-07

**Summary:**

AnaFP proposes an analytical approach to DNN fingerprinting that derives theoretical bounds on a stretch factor controlling fingerprint-to-boundary distance. The method formalizes robustness and uniqueness constraints to define an admissible interval, with a practical implementation using surrogate pools and quantile-based relaxation. Reviewers generally found the formulation clear and empirical results strong, but raised concerns about computational cost and memory, missing baselines, and robustness to adaptive or open-world scenarios.

**Reviewer Concerns:**

The rebuttal addressed several key issues. The authors clarified that computational steps are sequential rather than multiplicative and provided runtime profiling. They added comparisons with GMFIP and ADV-TRA showing consistently best AUC, evaluated stronger attacks like prune-KD, and provided detailed KD detection metrics. The distinction between small surrogate pools for bound estimation versus larger evaluation sets was clarified.

Outstanding concerns include high peak memory (up to 40GB for ViT-S/16), which limits scalability. The adaptive attack discussion relies on intuition rather than formal guarantees and the quantile selection remains heuristic despite empirical stability. The authors acknowledge that heavily diverged transfer learning is a fundamental limitation of this fingerprinting method. ZKy4 specifically asked about local linearization, curvature regularization, and Lipschitz constraints, but no formal analysis or empirical evaluation of these specific attacks was provided. However, the author's rebuttal is convincing in that an accuracy degradation would ensue.

**Reviewer Scores:**

Reviewer 5yER gave a 6 and would likely maintain or increase to 8, as their main concerns about cost, baselines, and I believe KD attack details were addressed with new experiments. Reviewer JWiN also gave a 6 and would likely maintain that score. Reviewer ZKy4 gave a 4 and would likely increase to 6, as they acknowledged some concerns were addressed but they still view adaptive robustness and quantile justification as problematic. Reviewer UJXn gave an 8 and confirmed maintaining their positive score after the rebuttal.

---

### Decision · Program_Chairs · 2026-01-26

Accept (Poster)